# Mannose receptor-derived peptides neutralize pore-forming toxins and reduce inflammation and development of pneumococcal disease

Karthik Subramanian[1] , Federico Iovino[1], Vasiliki Tsikourkitoudi[1] , Padryk Merkl[1], Sultan Ahmed[2], Samuel B Berry[3], Marie-Stephanie Aschtgen[1], Mattias Svensson[3], Peter Bergman[2,4], Georgios A Sotiriou[1] & Birgitta Henriques-Normark[1,5,6,*]

## Abstract

Cholesterol-dependent cytolysins (CDCs) are essential virulence factors for many human pathogens like *Streptococcus pneumoniae* (pneumolysin, PLY), *Streptococcus pyogenes* (streptolysin O, SLO), and *Listeria monocytogenes* (Listeriolysin, LLO) and induce cytolysis and inflammation. Recently, we identified that pneumococcal PLY interacts with the mannose receptor (MRC-1) on specific immune cells thereby evoking an anti-inflammatory response at sublytic doses. Here, we identified the interaction sites between MRC-1 and CDCs using computational docking. We designed peptides from the CTLD4 domain of MRC-1 that binds to PLY, SLO, and LLO, respectively. *In vitro*, the peptides blocked CDC-induced cytolysis and inflammatory cytokine production by human macrophages. Also, they reduced PLY-induced damage of the epithelial barrier integrity as well as blocked bacterial invasion into the epithelium in a 3D lung tissue model. Pre-treatment of human DCs with peptides blocked bacterial uptake via MRC-1 and reduced intracellular bacterial survival by targeting bacteria to autophagosomes. In order to use the peptides for treatment *in vivo*, we developed calcium phosphate nanoparticles (CaP NPs) as peptide nanocarriers for intranasal delivery of peptides and enhanced bioactivity. Co-administration of peptide-loaded CaP NPs during infection improved survival and bacterial clearance in both zebrafish and mice models of pneumococcal infection. We suggest that MRC-1 peptides can be employed as adjunctive therapeutics with antibiotics to treat bacterial infections by countering the action of CDCs.

**Keywords** calcium phosphate nanoparticles; mannose receptor C type 1; pore-forming toxins; *Streptococcus pneumoniae*; toxin inhibitory peptides
**Subject Categories** Immunology; Microbiology, Virology & Host Pathogen Interaction

## Introduction

Bacterial infections are leading causes of mortality and morbidity worldwide, and the emergence of resistance to many antibiotics is a major threat to society (Fischbach & Walsh, 2009). A common characteristic of many pathogenic bacteria, including some that have evolved drug resistance, is that they employ pore-forming toxins (PFTs) as virulence factors (Dal Peraro & van der Goot, 2016). PFTs constitute more than one-third of all cytotoxic toxins, making them the largest category of bacterial virulence factors (Gonzalez *et al*, 2008). Cholesterol-dependent cytolysins (CDCs) are a subclass of ß-PFTs that bind to cholesterol on eukaryotic cells and form barrel-shaped pores to mediate cytolysis (Los *et al*, 2013). CDCs promote bacterial virulence in many ways such as (i) induction of epithelial barrier dysfunction (Witzenrath *et al*, 2006), (ii) lysis of phagocytic immune cells (Domon *et al*, 2016), and (iii) facilitating bacterial invasion of host cells and intracellular survival (Subramanian *et al*, 2019b) (Birmingham *et al*, 2008). Prominent examples of bacterial CDCs include pneumolysin (PLY) from *Streptococcus pneumoniae*, streptolysin O (SLO) from *Streptococcus pyogenes*, listeriolysin O (LLO) from *Listeria monocytogenes*, intermedilysin (ILY) from *Streptococcus intermedius*, anthrolysin O (ALO) from *Bacillus anthracis,* and perfringolysin O (PFO) from *Clostridium perfringens* (Gilbert *et al*, 1999; Park *et al*, 2004; Boyd *et al*, 2016; Savinov & Heuck, 2017; Nguyen *et al*, 2019; Ogasawara *et al*, 2019; Subramanian *et al*, 2019a,b; Vogele *et al*, 2019).

1 Department of Microbiology, Tumor and Cell Biology, Karolinska Institutet, Stockholm, Sweden
2 Department of Laboratory Medicine, Karolinska Institutet, Huddinge, Sweden
3 Department of Medicine, Center for Infectious Medicine, Karolinska Institutet, Karolinska University Hospital, Huddinge, Sweden
4 The Immunodeficiency Unit, Department of Infectious Diseases, Karolinska University Hospital, Stockholm, Sweden
5 Clinical Microbiology, Karolinska University Hospital, Stockholm, Sweden
6 Lee Kong Chian School of Medicine (LKC) and Singapore Centre on Environmental Life Sciences Engineering (SCELSE), Nanyang Technological University, Singapore, Singapore
*Corresponding author. Tel: +46 852480000; E-mail: birgitta.henriques@ki.se

The structures of CDCs are conserved, consisting of four domains, and domain 4 is known to bind to the eukaryotic cell membrane (Tweten, 2005). Specifically, the highly conserved tryptophan-rich undecapeptide loop in domain 4 has been shown to bind cholesterol on eukaryotic membranes (van Pee *et al*, 2017). The binding triggers oligomerization of membrane-bound monomeric toxins into pre-pore structure. Conformational change triggers two α-helices in domain 3 to unfold into ß-hairpins which then insert into the membrane to form 250–300 Å pores. Due to their ubiquitous expression in bacterial pathogens, CDCs are attractive targets for development of novel broadly applicable antimicrobial therapeutics. The application of antibiotics to treat bacteremic patients is known to cause release of CDCs from lysed bacteria (Spreer *et al*, 2003), and hence adjunctive therapies to ameliorate the tissue damage caused by the released toxins are needed.

In our recent study, we identified that the human mannose receptor (MRC-1/CD206) expressed on dendritic cells and lung alveolar macrophages binds to PLY at sublytic doses. MRC-1 is an endocytic receptor that binds and internalizes glycoproteins with terminal mannose, fucose, or N-acetylglucosamine residues, as well as pathogens bearing high mannose structures. However, at sublytic doses, MRC-1 binds to PLY, independent of the capsular polysaccharides, resulting in an anti-inflammatory response and enhanced intracellular survival of pneumococci (Subramanian *et al*, 2019b).

Here, we used computational docking to predict the sites of interaction between MRC-1 and CDCs. By constructing peptides from the region of interaction, we studied whether we may inhibit the action of CDCs. Using human macrophages, DCs, and 3D lung epithelial tissue models, we investigated the effect of treatment with the peptides on toxin-induced cytolysis, epithelial damage, inflammation, and bacterial invasion. Finally, we explored the use of biocompatible calcium phosphate nanoparticles (CaP NPs) as peptide nanocarriers for intranasal delivery of the peptides to the lungs and studied their effect on survival and bacterial clearance using two *in vivo* infection models, mice and zebrafish.

# Results

## MRC-1 co-localizes with the bacterial CDCs, PLY, LLO, and SLO, in human dendritic cells

To test whether MRC-1 could serve as a common receptor for structurally conserved bacterial CDCs, we incubated human monocyte-derived DCs with a non-cytolytic dose (0.2 μg/ml) of the purified toxins, PLY, LLO, and SLO, for 45 min, and performed immunofluorescence staining. We found that MRC-1 co-localized with all the three CDCs in DCs (Fig 1A–C). To verify uptake by DCs, we co-stained for the early endosomal antigen, EEA-1, and found that MRC-1 co-localized with the three CDCs along with EEA-1 (Fig 1A–C). The extent of co-localization of MRC-1 is quantified in Fig EV1A. To test whether the cholesterol-binding loop in domain 4 of the CDCs, that binds to the host cell membrane, is also involved in the interaction with MRC-1, we used toxoid derivates of PLY (W433F) and LLO (W489F) bearing point mutations in a key tryptophan residue of the cholesterol-binding loop. In contrast to the wild-type toxins, the toxoids showed drastically reduced binding to the DCs and did not co-localize with MRC-1, indicating that the cholesterol-

binding loop of PLY and LLO is essential for the interaction with MRC-1 on DCs (Fig EV1B and C). Further, as shown previously for PLY (Subramanian *et al*, 2019b), antibody blockade of MRC-1 also impaired the binding of LLO and SLO in DCs (Fig EV1D and E) and impaired co-localization with EEA-1 (Fig EV1F).

## The CTLD4 domain of MRC-1 interacts with the cholesterol-binding loop of bacterial CDCs

Next, we wanted to determine the specific sites of interaction between MRC-1 and the CDCs. We recently showed that the CTLD4 domain of MRC-1 interacts with the membrane-binding domain 4 of PLY (Subramanian *et al*, 2019b). Hence, we performed computational docking of the crystal structures of PLY, LLO, and SLO with the CTLD4 domain of MRC-1 on the ClusPro 2.0 docking server (Kozakov *et al*, 2017), based on least energy configuration. The structures of PLY, LLO, and SLO are conserved and consist of four domains, D1–D4, wherein domain 4 binds to cholesterol on the eukaryotic cell membrane (Fig 2A–C). The tryptophan-rich undecapeptide loop (highlighted in yellow) in domain 4 binds to the cell membrane and is highly conserved among the CDCs. The cholesterol-binding loop has been shown to alter the avidity of cholesterol binding by altering oligomerization (Dowd *et al*, 2012). Modeling results suggest that the CTLD4 of MRC-1 (red) interacts with the cholesterol-binding loop (yellow) in domain 4 of PLY, LLO, and SLO, respectively (Fig 2A–C). Particularly, modeling predicted that the tryptophan residues, W433 of PLY and W489 of LLO, located in the cholesterol-binding loop of domain 4, are involved in hydrogen bonding interactions with CTLD4 of MRC-1. This is in line with our earlier data, where we observed that the mutant derivatives, PLY (W433F) and LLO (W489F), did not interact with MRC-1 (Fig EV1B and C).

## MRC-1 peptides bind to bacterial CDCs and inhibit their induction of hemolysis and cytolysis of macrophages

Since our results indicated that MRC-1 interacts with the cholesterol-binding loop of the CDCs, we hypothesized that peptides from the region of interaction could inhibit toxin interactions with eukaryotic cells. Therefore, we constructed overlapping 13-mer peptides from the CTLD4 of MRC-1, and two negative control peptides from the fibronectin type II domain and intracellular tail (Fig EV2A and Appendix Table S1). The peptides were commercially synthesized to > 95% purity and screened for their ability to inhibit hemolysis of red blood cells induced by purified bacterial toxins. Addition of peptides P1-P6 inhibited PLY- and LLO-induced hemolysis as opposed to the control peptides, CP1 and CP2, as evident by the residual red blood cell pellet at the end of the hemolytic assay (Fig EV2B). Peptides P2 and P3 were the most potent, inhibiting hemolysis by up to 50% (Fig EV2C). In agreement, both peptides P2 and P3 contain a continuous stretch of ≥ 3 amino acids that formed hydrogen bonding interactions with the amino acids in the cholesterol-binding loop of the CDCs. The amino acids involved in hydrogen bonding interactions are indicated in Appendix Table S2. We found that both peptides, P2 and P3, were surface localized on the MRC-1 CTLD4 domain, which is ideal for interactions with bacterial toxins (Fig 2D and E). Cholesterol, a known inhibitor of PLY-induced hemolysis, was used as a positive

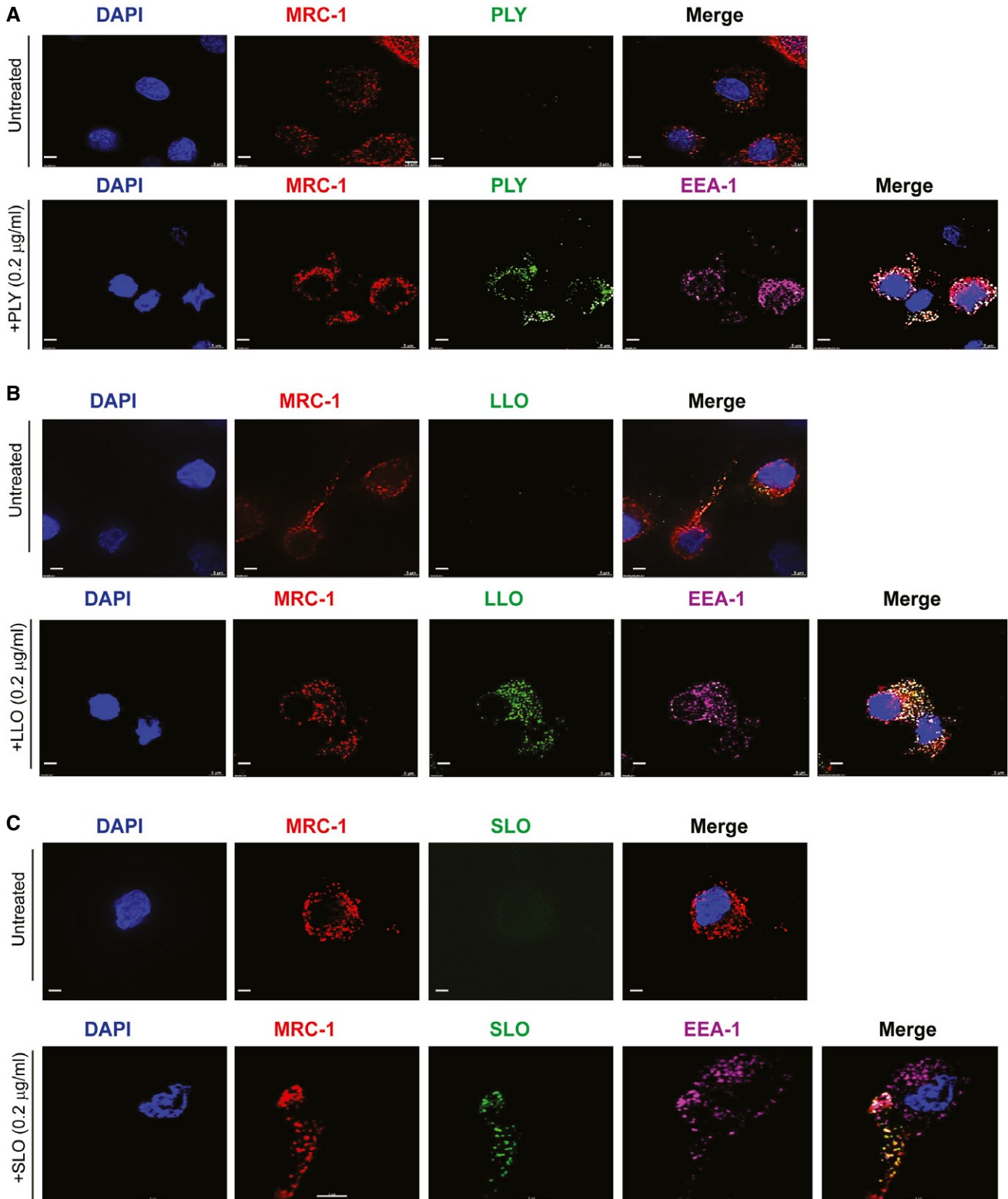

**Figure 1.  MRC-1 co-localizes with the bacterial CDCs, PLY, LLO, and SLO, in human dendritic cells.**

A–C   Human DCs were incubated with a non-cytolytic dose (0.2 µg/ml) of purified (A) PLY, (B) LLO, or (C) SLO for 45 min. Immunofluorescence staining shows that PLY, LLO, and SLO (green) co-localize with MRC-1 (red) in DCs and EEA-1 (purple, early endosomes). All scale bars, 5 µm. Images are representative of three independent experiments.

Source data are available online for this figure.

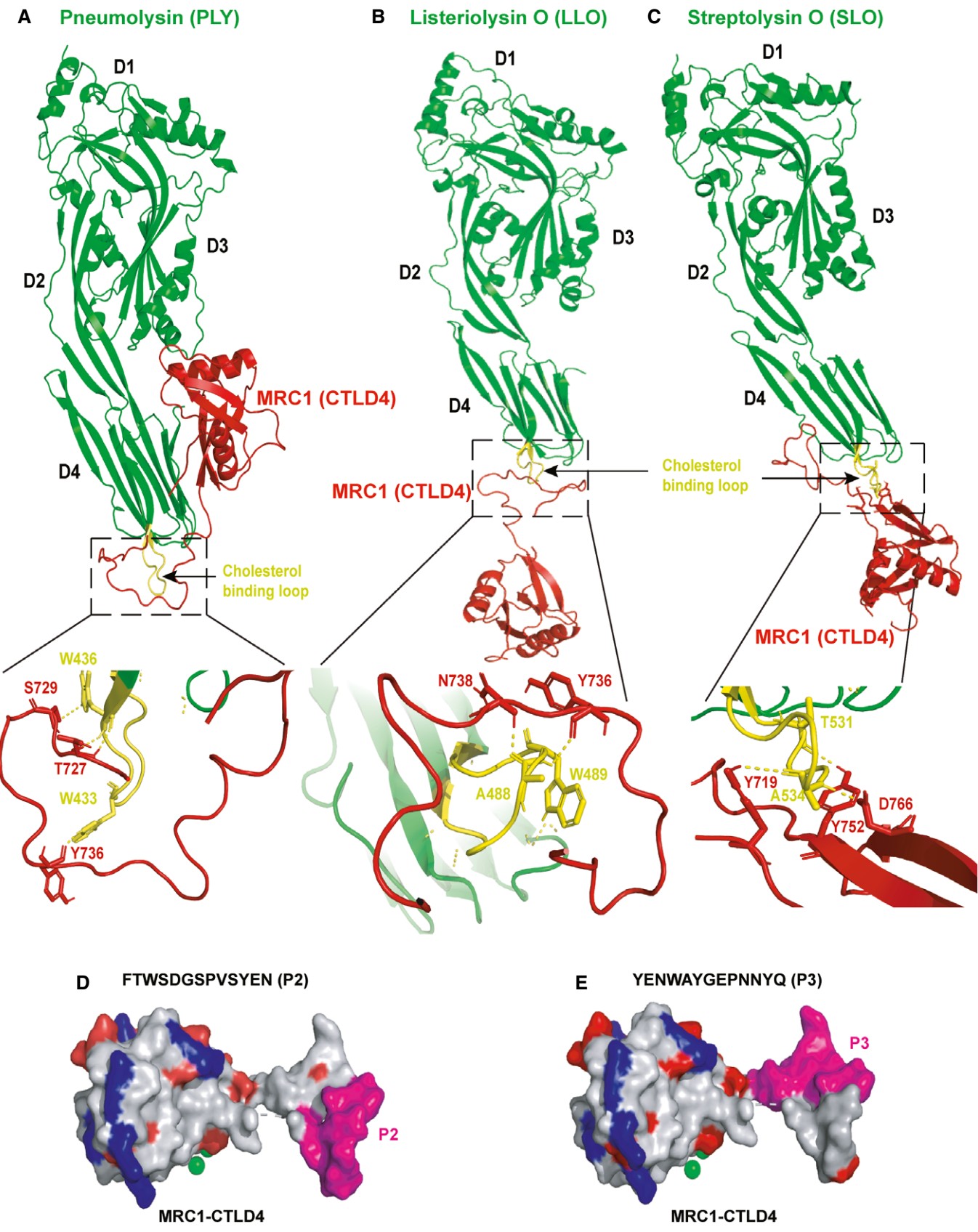

**A** Pneumolysin (PLY)

**B** Listeriolysin O (LLO)

**C** Streptolysin O (SLO)

**D** FTWSDGSPVSYEN (P2)

**E** YENWAYGEPNNYQ (P3)

**Figure 2.**

**Figure 2. The CTLD4 domain of MRC-1 interacts with the cholesterol-binding loop of bacterial CDCs.**

A–C  Computational docking of (A) PLY, (B) LLO, and (C) SLO (in green) with the CTLD4 domain of MRC-1 (in red) was performed using the ClusPro 2.0 docking server. Modeling based on least energy configurations indicate that the unstructured loop of MRC-1 docks to the conserved cholesterol-binding loop (in yellow) of PLY, LLO, and SLO. The amino acid residues predicted to involve in hydrogen bonding interactions are zoomed in below.

D, E  3D view of the CTLD4 domain of MRC-1 showing the surface location of the peptides P2 and P3 (in pink). Acidic and basic amino acid residues are shown in red and blue, respectively. Indicated in green is the calcium binding site.

Source data are available online for this figure.

control. Bovine serum albumin (BSA) was used as a negative control to verify that the inhibition of PLY-mediated hemolysis by the MRC-1 peptides was specific.

Next, we set up an ELISA to ascertain the binding of MRC-1 peptides to the purified CDCs and focused on peptides P2 and P3 that showed the highest potency. Increasing doses of purified PLY or BSA (negative control) were added to peptides P2, P3, or control peptides CP1 or CP2, that were immobilized on the plates. Peptides P2 and P3 were found to bind dose-dependently to PLY, but not to BSA, suggesting that the binding was specific (Fig 3A). The control peptides, CP1 and CP2, showed only background levels of binding. Peptides P2 and P3 also bound dose-dependently to LLO and SLO (Fig EV3A and B). To determine the optimal working concentration of the peptides, we performed a hemolysis assay by titrating increasing concentrations of the peptides in the presence of 1 µg/ml PLY. To verify that the peptide sequence rather than its amino acid composition is crucial for inhibiting toxin activity, we used a scrambled version of peptide P2 as a control. We found that both peptides P2 and P3 dose-dependently inhibited hemolysis induced by the purified toxins PLY, LLO, and SLO, resulting in up to 50% inhibition at concentrations ranging from 10 to 100 µM (Figs 3B, and EV3C and D). However, the scrambled peptide P2 did not have any significant effect on hemolysis. Using regression analysis, we determined the ED50 of the peptides P2 and P3 against the purified toxins (Appendix Table S3).

To visualize inhibition of the bacterial CDCs by the peptides in real time, we performed live imaging using human THP-1 monocyte-derived macrophages upon addition of PLY in the presence or absence of peptide P2. The cells were pre-loaded with the live-dead stain comprising of Calcein AM and propidium iodide that differentially stain live and dead cells green and red, respectively. The cells were imaged for 20 min post-addition of 0.5 µg/ml PLY in the presence or absence of 100 µM peptide P2, or control peptide CP2. This dosage of PLY was chosen since at concentrations between 0.5 and 1 µg/ml PLY has been shown to activate the inflammasome and induce acute lung injury in acute pneumonia model (Stringaris et al, 2002; Witzenrath et al, 2006). In contrast to untreated cells, addition of PLY resulted in membrane blebbing and positive staining of cells by propidium iodide (Movies EV1 and EV2). However, in the presence of the peptide P2, but not of the control peptide CP2, most of the cells were intact and stained green, indicating protection from cytolysis (Movies EV3 and EV4). Cholesterol was used as positive control (Movie EV5) and BSA as a negative control to show the specificity of the peptides (Movie EV6). No cell death was observed when the toxoid form of PLY, Pdb (W433F), that is defective in pore-formation, was added (Movie EV7). Peptide P2 also protected the cells from cytolysis mediated by LLO and SLO (Movies EV8–EV11). To quantify cell death, we measured the release of lactate dehydrogenase (LDH) from lysed cells into the culture supernatant.

Peptides P2 and P3 significantly reduced cell death of macrophages induced by PLY, LLO, or SLO (Fig 3C), but no significant effect was observed with the controls, scrambled peptide P2 and peptide CP2.

Then, we tested whether the peptides could bind in situ to toxin-producing bacteria. We incubated FITC-labeled peptides with the PLY-producing pneumococcal strain T4 (TIGR4 of serotype 4) and its isogenic PLY mutant, T4Δply, and analyzed by fluorescence microscopy. We found that peptide P2 bound to T4, but not to the PLY-deficient strain (Fig 3D), suggesting that the peptides bind to PLY directly on the surface of the bacteria. However, the control peptide CP2 did not show any binding, indicating that the binding was specific. Moreover, we measured the hemolytic activity of toxin-producing bacteria in the presence of 100 µM peptide P2 or CP2. We found that in the presence of peptide P2, the hemolytic activity of the pneumococcal strain T4 was significantly reduced, while the control peptide CP2 showed no effect on the hemolytic activity (Fig 3E). Peptide P2 also significantly reduced the hemolytic activity of the strains S. pyogenes type M1T1 and L. monocytogenes type 1 that express the toxins SLO and LLO, respectively (Fig EV3E and F).

## MRC-1 peptides reduce pro-inflammatory responses in macrophages and cytotoxicity in a 3D lung tissue model

Bacterial CDCs, such as PLY, are known to induce a robust inflammatory response in human macrophages (McNeela et al, 2010). Hence, we next measured the release of pro-inflammatory cytokines by human THP-1-derived macrophages at 18 h post-challenge with PLY, LLO, or SLO (0.5 µg/ml) in the presence of 100 µM peptides P2 or P3, or control peptide CP2. We found that in contrast to CP2, peptides P2 and P3 significantly reduced the release of the chemokine IL-8 and the pro-inflammatory cytokines TNF-α and IL-12 (Figs 4A, and EV4A and B). Cholesterol was used as positive control.

To study toxin-mediated tissue pathology associated with pneumococcal pneumonia, we utilized a green-fluorescent protein (GFP)-expressing 3D lung epithelial tissue model with an air-exposed stratified epithelial layer on top of a lung fibroblast matrix layer (Fig 4B). The fibroblasts and epithelial cells in the model were derived from human lung tissue and have been shown to mimic the lung physiological conditions like epithelial stratification, cilia, and mucus secretion (Nguyen Hoang et al, 2014; Mairpady Shambat et al, 2015). We performed live imaging to monitor the loss of GFP expression by the epithelial cells upon stimulation with 1 µg/ml PLY in the presence or absence of 100 µM peptide P2 or control peptide CP2. We found that at 3 h post-challenge, PLY stimulation led to a greater reduction of the GFP signal compared to the untreated control, indicating epithelial disruption (Fig 4C). The results were quantified in Fig EV4C. The loss of GFP expression was

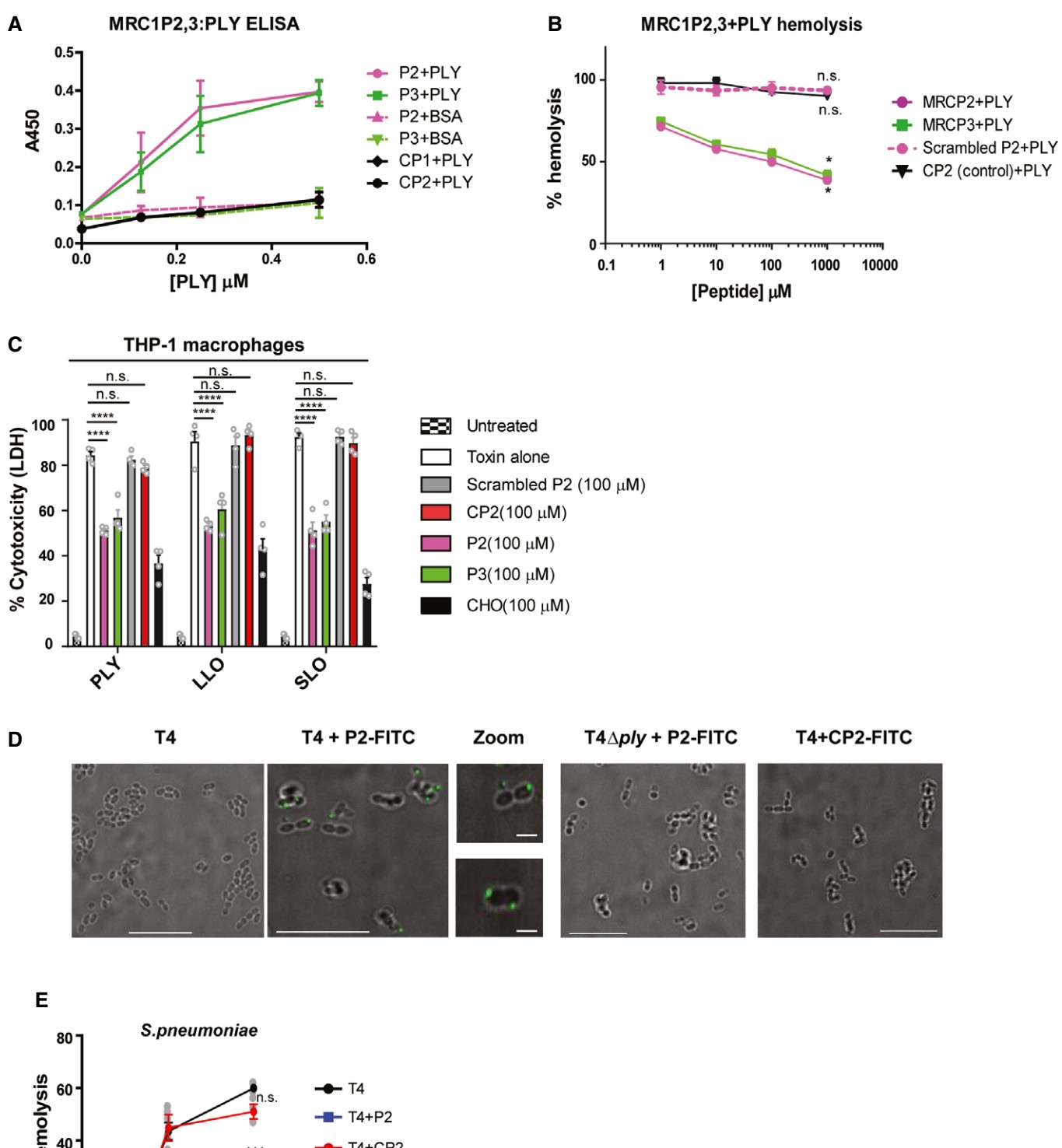

**Figure 3.**

**Figure 3.  MRC-1 peptides bind to bacterial CDCs and inhibit their induction of hemolysis and cytolysis of macrophages.**

A  ELISA showing the dose-dependent binding of plate-bound MRC-1 peptides P2, P3, and the control peptides CP1 and CP2 to PLY (0–0.5 µM). BSA was used as negative control to show the binding specificity. Data are mean ± s.e.m. of two independent experiments, each containing three replicates per condition.

B  Hemolysis assay (n = 4) of 1 µg/ml purified PLY in the presence of increasing concentrations of MRC-1 peptides, P2, scrambled P2, P3, and control peptide CP2 (1–1000 µM). Data represent mean ± s.e.m. *P < 0.05 by one-way ANOVA with Dunnett's *post hoc* test for multiple comparisons. Exact P values are shown in Appendix Table S4.

C  LDH cytotoxicity assay in human THP-1 macrophages stimulated with purified PLY, LLO, or SLO (0.5 µg/ml) in the presence or absence of 100 µM peptides P2, scrambled P2, P3, or control peptide CP2 for 18 h. Cholesterol (100 µM) was used as positive control to inhibit hemolysis. Data are mean ± s.e.m. from 4 independent experiments. ****P < 0.0001 by two-way ANOVA with Bonferroni *post hoc* test for multiple comparisons. n.s. denotes not significant. Exact P values are shown in Appendix Table S4.

D  Binding of FITC-labeled peptides P2 and CP2 to wild-type pneumococci, TIGR4 (T4), and its isogenic PLY mutant (T4Δ*ply*) was visualized by fluorescence microscopy. Scale bars, 10 µm. In magnified images, scale bars, 1 µm. Images are representative of three independent experiments.

E  The hemolytic activity of wild-type pneumococci, TIGR4 (T4) and the PLY mutant, T4Δ*ply* in the presence of 100 µM peptide P2 and CP2. Data are the mean ± s.e.m. of three independent experiments. ***P < 0.001 by one-way ANOVA with Bonferroni *post hoc* test for multiple comparisons. n.s. denotes not significant. Exact P values are shown in Appendix Table S4.

Source data are available online for this figure.

significantly lower in the presence of peptide P2 in comparison with the control peptide CP2. Cholesterol was used as positive control to inhibit epithelial damage by PLY. To quantify epithelial damage, we measured LDH release into the supernatant at 18 h post-challenge. Concurrent with the loss of GFP expression, PLY stimulation of the lung tissue model induced a robust release of LDH into the supernatant, which was significantly reduced in the presence of the peptide P2 (Fig EV4D). The control peptide CP2 did not show any significant reduction in LDH release compared to the PLY-treated model.

Epithelial cells in the respiratory tract constitute the primary barrier against pathogens and mediate innate immune response by producing antibacterial factors as well as secrete pro-inflammatory cytokines and chemokines to attract phagocytic cells to the site of infection. Excessive inflammatory responses cause tissue damage and thereby increase mortality of respiratory infections. Hence, we measured the pro-inflammatory cytokine release in the lung tissue model in response to PLY with or without the peptides. We found a significantly reduced release of the neutrophil chemokine IL-8 and of TNF-α upon addition of peptide P2, but not with peptide CP2 (Fig EV4E and F).

**Toxin-mediated bacterial invasion of the lung epithelium and intracellular bacterial survival are reduced by treatment with MRC-1 peptides**

During invasive diseases such as pneumonia, bacteria breach the tight junctions of the epithelial barrier in order to invade underlying tissues and spread to sterile sites via the bloodstream. We therefore investigated the role of CDCs in promoting bacterial invasion into the epithelium by counting the colony-forming units (CFUs) upon infection with strain T4 or its isogenic PLY mutant (T4Δ*ply*) in our 3D lung epithelial model. We found that the PLY mutant strain showed reduced invasion into the epithelium as compared to the wild-type T4 (Fig 4D), suggesting that PLY promotes pneumococcal invasion of the epithelium. Addition of peptide P2, but not of CP2, significantly reduced the number of invading bacteria, implying that the peptide inhibits bacterial translocation across the epithelium by blocking PLY (Fig 4D). The anti-PLY antibody was used as a positive control to demonstrate the role of PLY in pneumococcal invasion.

Besides cytolysis, PLY also promotes intracellular survival of pneumococci in DCs and lung alveolar macrophages (Subramanian

*et al*, 2019b), as well as trafficking across the blood brain barrier to invade the brain (Surve *et al*, 2018). Hence, we measured the intracellular bacterial load of pneumococcal strains T4 of serotype 4 and D39 of serotype 2 in DCs, in the presence or absence of the PLY-binding peptide P2 at 3 h post-infection. Addition of peptide P2 significantly reduced the number of intracellular bacteria in DCs (Fig 4E), while the control peptide CP2 did not show any significant difference. The anti-PLY antibody was used as a control to verify the effect of PLY on intracellular pneumococcal survival. The actin polymerization inhibitor, cytochalasin D (CytD), was used as positive control to inhibit DC phagocytosis. The inhibition of bacterial invasion by the peptides into the lung epithelium and DCs (Fig 4D and E) could be due to reduced bacterial entry as well as intracellular survival. We also performed fluorescence microscopy to visualize intracellular bacteria in DCs in the presence of peptides P2 and CP2. The results agreed with the data from the CFU plating assay. Intracellular pneumococci (green) were detected in infected DCs that co-localized with MRC-1 (red) (Fig EV4G). DCs that were treated with peptide P2 or anti-PLY were devoid of intracellular bacteria.

A key strategy of phagocytic immune cells to eliminate intracellular bacteria is through autophagy. During autophagy, the bacteria are enclosed by double-membrane structure called phagophore. The autophagy protein microtubule-associated 1 light chain 3 (LC3) undergoes cleavage and associates with the phagophore upon lipidation (Siqueira *et al*, 2018). The autophagosomes fuse with lysosomes leading to degradation of intracellular bacteria. The mannose receptor, MRC-1, has been implicated in inhibiting phagosome maturation as well as fusion with lysosomes (Sweet *et al*, 2010; Rajaram *et al*, 2017; Subramanian *et al*, 2019b). We therefore tested the effect of the PLY-inhibiting peptides on the fate of intracellular pneumococci by immunofluorescence microscopy. DCs were infected with the unencapsulated strain T4R (to get better uptake than with encapsulated bacteria) or its isogenic PLY mutant T4RΔ*ply* and were immunostained for MRC-1, pneumococci (antiserum), and the autophagy marker LC3B, at 3 h post-infection. We found that intracellular T4R bacteria in DCs co-localized with MRC-1 but did not co-stain with the autophagy marker LC3B (Fig 4F and G). This was in contrast to T4RΔ*ply* which co-localized with LC3B (Figs 4G and EV5). In DCs treated with peptide P2, intracellular T4R did not co-localize with MRC-1, but co-stained with LC3B, suggesting that the bacteria are targeted for degradation by autophagy. In

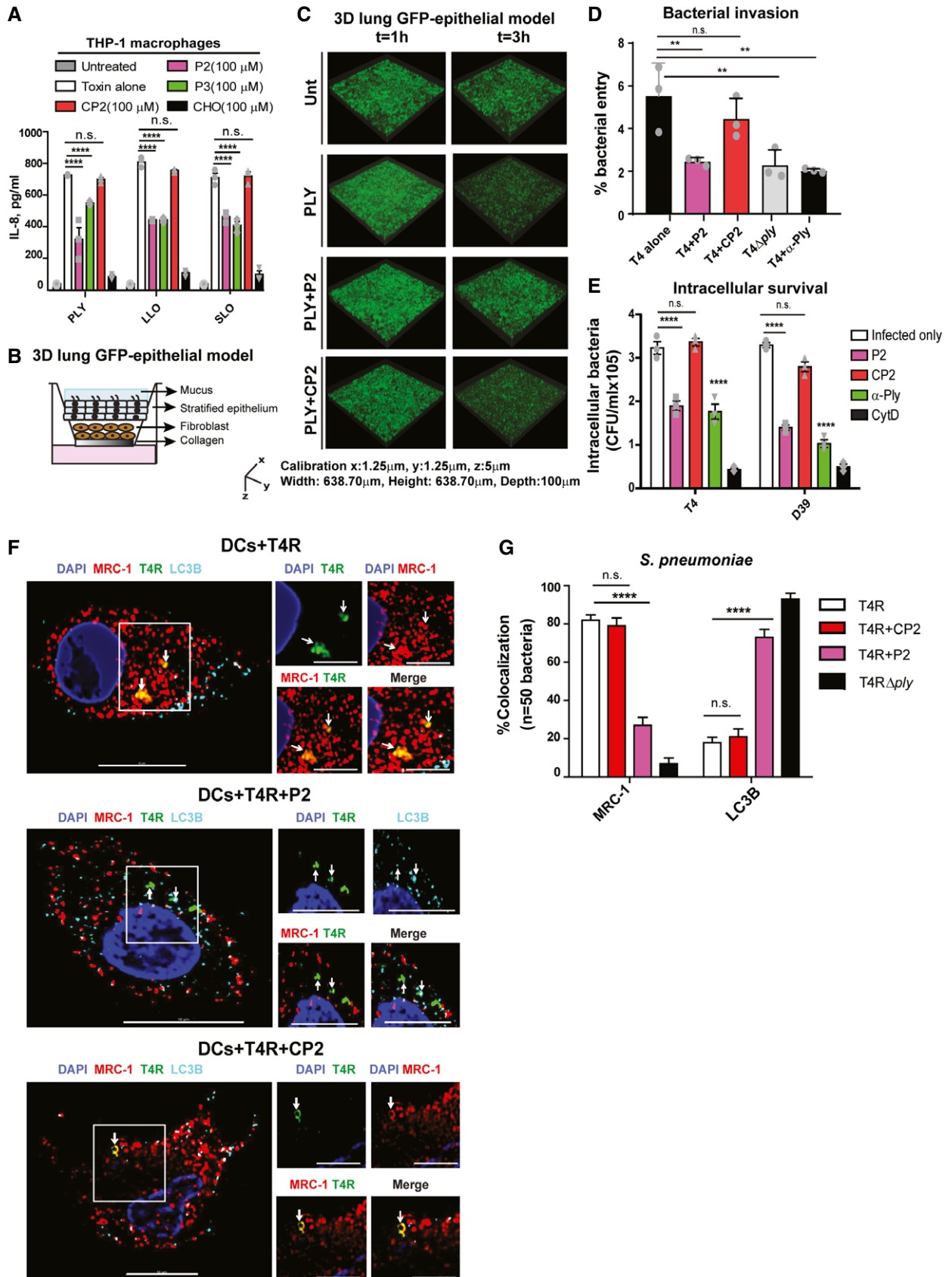

**Figure 4.**

Figure 4.   PLY-mediated bacterial invasion of the lung epithelium and intracellular bacterial survival are inhibited by MRC-1 peptides.

A    IL-8 released by human THP-1 macrophages stimulated with purified PLY, LLO, or SLO (0.5 μg/ml) in the presence or absence of 100 μM peptides P2, P3, or control peptide CP2 for 18 h. Cholesterol (100 μM) was used as positive control to inhibit hemolysis. Data are mean ± s.e.m. from three independent experiments. ****$P < 0.0001$ by two-way ANOVA with Bonferroni post hoc test for multiple comparisons. n.s. denotes not significant. Exact $P$ values are shown in Appendix Table S4.
B    Schematic showing the cellular architecture of the 3D lung epithelial model.
C    3D volume images of the GFP-lung epithelial models at 1 and 3 h post-stimulation with 1 μg/ml PLY in the presence or absence of 100 μM peptide P2 or the control peptide CP2. Images are representative of two independent experiments with $n = 3$ models/condition.
D    Invasion of wild-type pneumococci T4 (TIGR4) or its isogenic PLY mutant T4Δply into the lung epithelial models ($n = 3$/condition) in the presence or absence of 100 μM peptide P2 or the control peptide CP2 at 2 h post-infection was measured using CFU viability assay following gentamicin killing of extracellular bacteria. Anti-PLY was used as control to test the effect of blocking PLY. Data are mean ± s.e.m. of $n = 3$ models/condition from two independent experiments. % bacterial entry = (bacteria uptaken/input) × 100. **$P < 0.01$ by one-way ANOVA with Dunnett's post hoc test for multiple comparisons. n.s. denotes not significant. Exact $P$ values are shown in Appendix Table S4.
E    Human DCs were infected with type 4 and type 2 pneumococci, T4 and D39, respectively, at MOI of 10 in the presence or absence of 100 μM peptides, P2 or CP2, and intracellular bacteria were counted at 3 h post-infection following gentamicin killing of extracellular bacteria. Cytochalasin D (0.5 mM) was used as negative control to inhibit phagocytosis. Anti-PLY was used as control to test the effect of blocking PLY. Data are mean ± s.e.m. of three independent experiments. ****$P < 0.0001$ by two-way ANOVA with Bonferroni post hoc test for multiple comparisons. n.s. denotes not significant. Exact $P$ values are shown in Appendix Table S4.
F    DCs were infected with the unencapsulated pneumococcal strain T4R in the presence of 100 μM peptides, P2 or CP2 at MOI of 10 for 2 h. Immunofluorescence microscopy images show that in DCs treated with peptide P2 (but not the control peptide CP2), intracellular T4R (green) do not co-localize with MRC-1 (red), but with the autophagy protein LC3B (cyan). Images are representative of three independent experiments. Scale bars, 10 μm. In magnified images, scale bars, 5 μm. Arrows indicate regions of co-localization of intracellular T4R with MRC-1 and LC3B.
G    Quantification of percentage of intracellular S. pneumoniae ($n = 50$) in infected DCs that co-localize with MRC-1 and LC3B. Data are mean ± s.e.m. from two independent experiments. ****$P < 0.0001$ by two-way ANOVA with Bonferroni post hoc test for multiple comparisons. n.s. denotes not significant. $P$ values are shown in Appendix Table S4.

Source data are available online for this figure.

contrast to P2, treatment with the control peptide CP2 did not promote co-localization of intracellular bacteria with LC3B. We also examined the intracellular fate of SLO-producing *S. pyogenes* type M1T1, and its isogenic SLO mutant. The results were consistent with those for *S. pneumoniae,* showing that the SLO-producing strain co-localized with MRC-1, but not with LC3B. Upon addition of peptide P2, intracellular bacteria co-stained with LC3B (Appendix Fig S1A and B). The SLO mutant strain consistently co-stained with LC3B, irrespective of peptide treatment.

**Treatment with peptide P2 reduces development of pneumococcal disease *in vivo***

To assess the therapeutic potential of peptide P2 against pneumococcal infection *in vivo*, we first used a zebrafish (*Danio renio*) embryo model in which pneumococci were microinjected into the yolk sac of fertilized embryos. Zebrafishes have been shown to be useful models to study infectious diseases (Sullivan *et al*, 2017) owing to their optical transparency, ease of high-throughput screening, and conservation of the major components of the human immune system (Lam *et al*, 2004; Renshaw & Trede, 2012). Also, they have been used to study pneumococcal pathogenesis (Saralahti *et al*, 2014; Jim *et al*, 2016). Thus, we infected zebrafish embryos 3–4 h post-fertilization with 500 CFU of wild-type T4 or its isogenic PLY mutant, T4Δ*ply*. We found that embryos infected with T4Δ*ply* showed significantly higher survival in comparison with the wild-type T4 strain (Fig 5A), confirming the importance of PLY in pneumococcal virulence. Importantly, co-administration of peptide P2 (1 nM) during T4 infection significantly improved the survival compared to infection alone (Fig 5B). In contrast, the control peptide CP2, that does not bind PLY, did not show any significant effect on the survival of infected zebrafishes, confirming the specificity of peptide P2.

Although peptides are specific, they have a limited stability and bioavailability *in vivo* that needs to be improved in order for them to be used as therapeutics (Lau & Dunn, 2018). Therefore, we used

biocompatible calcium phosphate (CaP) nanoparticles (NPs) as peptide nanocarriers to minimize degradation and to achieve rapid targeting to the lungs (Tsikourkitoudi *et al*, 2020). Peptide conjugation to nanoparticles also allows for their non-invasive delivery by inhalation for rapid targeting to the lungs rather than the conventional intravenous route. Thus, the MRC-1 peptide, P2, and control peptide, CP2, were loaded onto CaP NPs by physisorption upon overnight incubation as described previously (Tonigold *et al*, 2018). The morphology and size distribution of the NPs for peptide loading were characterized using transmission electron microscopy (Appendix Fig S2A) and dynamic light scattering (Appendix Fig S2B). The particles exhibited the characteristic fractal-like agglomerate nanostructure (Sauter mean diameter ~ 8 nm) and the mean hydrodynamic size of the peptide-loaded NPs was ~ 770 nm when suspended in PBS. The amount of peptide loaded onto CaP NPs was quantified using the BCA assay (Appendix Fig S2C). Up to 75–150 mg peptide/g CaP could be loaded at a particle concentration of 250 μg/ml. Further, the peptide activity upon conjugation to NPs was verified by hemolysis assay and a dose-dependent inhibition of T4-induced hemolysis by peptide P2-loaded nanoparticles (P2-NPs) was observed, but not by nanoparticles conjugated with control peptide CP2 (CP2-NPs) (Appendix Fig S2D). The unloaded NPs alone did not induce any significant hemolysis (Appendix Fig S2D). Importantly, we found that co-injection of zebrafish embryos with P2-loaded NPs (1 nM) and strain T4 improved survival of the fishes compared to unloaded NPs and T4 alone (Fig 5B).

Next, we used an intranasal mouse pneumonia model to investigate the effects of the peptide-loaded NPs on bacterial virulence *in vivo*. First, we studied lung delivery and *in vivo* distribution of the peptide nanocarriers by administering Cy7-tagged P2-CaP NPs intranasally to mice and imaging the lungs post-mortem using the IVIS imaging at 1 and 24 h post-administration. Unloaded NPs and Cy7 dye alone were used as negative and positive controls, respectively. Images showed that P2-NPs were efficiently distributed in both lung lobes at 1 h and the peptide could be detected in the lungs

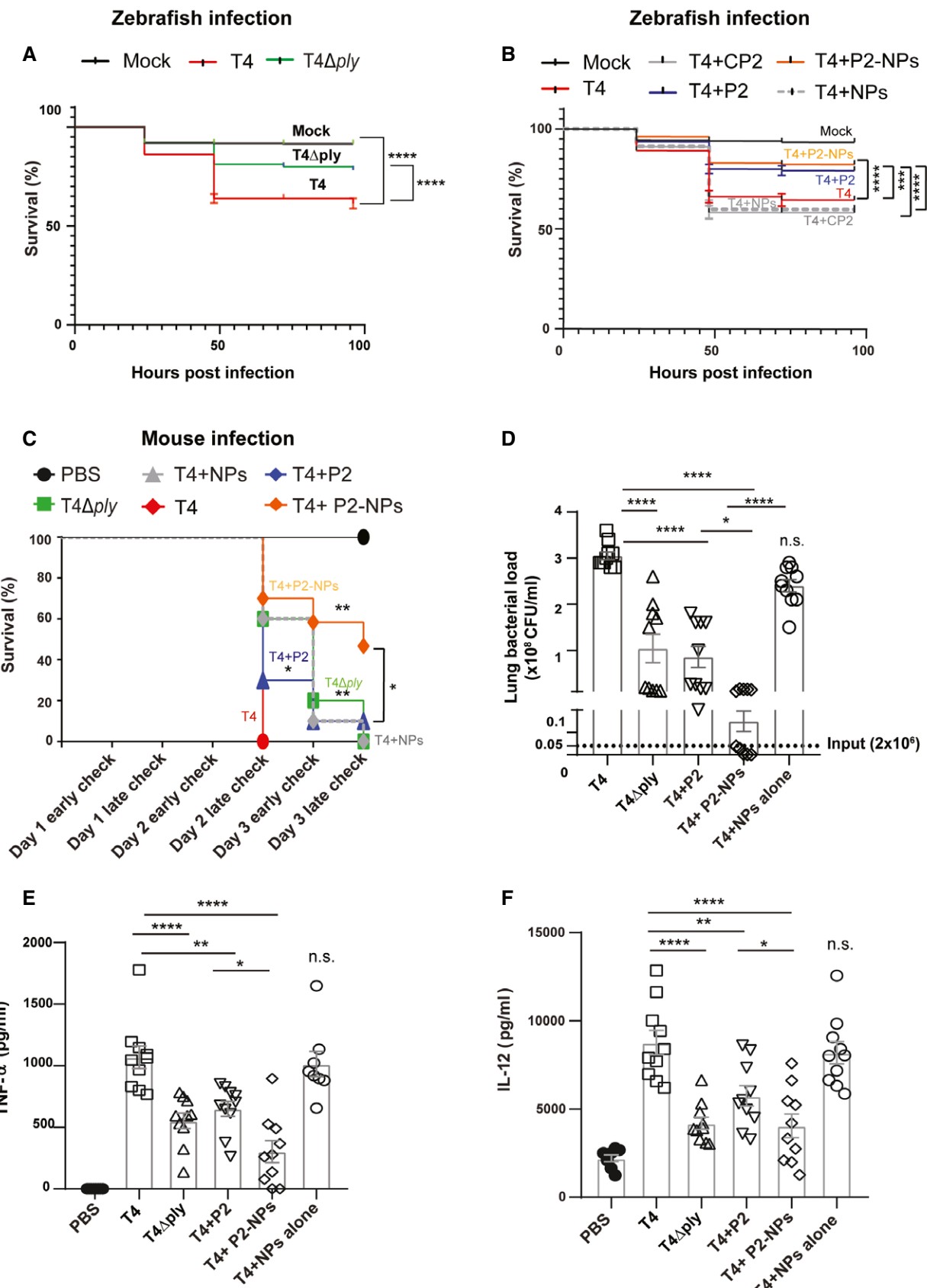

**Figure 5.**

**Figure 5.  Treatment with peptide P2 reduces development of pneumococcal disease *in vivo*.**

A    Survival percentage of 3–4 dpf zebrafish embryos ($n \geq 156$) upon infection with *S. pneumoniae* T4 alone or its isogenic PLY mutant, T4Δ*ply*. Injection with E3 growth medium served as mock control. ****$P < 0.0001$ by Mantel–Cox test. Exact *P* values are shown in Appendix Table S4.

B    Zebrafish survival percentage upon infection with T4 alone or together with peptide P2 or CP2 or P2-conjugated CaP NPs (P2-NPs). ***$P < 0.0005$ and ****$P < 0.0001$ by Mantel–Cox test. Exact *P* values are shown in Appendix Table S4.

C    Survival of mice ($n = 10$) upon intranasal infection with $2 \times 10^6$ CFU of *S. pneumoniae* T4 together with peptide P2 or CP2 or P2-NPs over 3 days post-infection. Infected mice were checked twice daily in the morning at 9 am (early check) and in evening at 7 pm (late check) for clinical symptoms. Unloaded NPs and the isogenic PLY mutant strain (T4Δ*ply*) served as negative controls. *$P < 0.05$ and **$P < 0.005$ by Mantel–Cox test. Exact *P* values are shown in Appendix Table S4.

D–F  Bacterial load (D) and levels of pro-inflammatory cytokines (E) TNF-α and (F) IL-12 in the lungs of infected mice ($n = 10$) were measured post-sacrifice. Data represent mean $\pm$ s.e.m. *$P < 0.05$, **$P < 0.01$, and ****$P < 0.0001$ by one-way ANOVA with Bonferroni's *post hoc* test for multiple comparisons. n.s. denotes not significant. Exact *P* values are shown in Appendix Table S4.

Source data are available online for this figure.

even at 24 h (Appendix Fig S2E). Peptide P2 or P2-NPs were not toxic as revealed by the lung histology analysis, which showed that the mice had a normal lung morphology with intact alveolar space and absence of inflammatory cells (Appendix Fig S2F). Moreover, these mice did not develop any clinical symptoms at 24 h post-administration. Subsequently, we challenged mice intranasally with $2 \times 10^6$ CFUs of *S. pneumoniae* T4 strain or the PLY mutant, T4Δ*ply*, combined with peptide P2 (5 μg/mouse) or peptide-conjugated nanoparticles (5 μg peptide; 25 μg CaP NPs/mouse) in 50 μl PBS. In agreement with the zebrafish data, mice infected with T4Δ*ply* showed higher survival compared to T4-infected mice (Fig 5C). Further, mice treated with peptide P2 showed prolonged survival in comparison with the infected group. Indeed, peptide P2-treated mice had similar survival as T4Δ*ply*-infected mice, indicating that the peptide effectively inhibited PLY *in vivo*. Importantly, administration of peptide P2-conjugated NPs resulted in a significantly higher survival (50%) at the end of the experiment in comparison with mice treated with blank NPs or only infected (Fig 5C). Also, P2-NP-treated mice showed higher survival in comparison with the free peptide, indicating that the NPs improved the peptide efficacy *in vivo*. In accordance with the higher survival, P2-NP-treated mice also had significantly reduced bacterial load in the lungs compared to mice infected only or challenged with blank NPs (Fig 5D). In addition, treatment with peptide P2 or P2-NPs significantly reduced the levels of inflammatory cytokines, TNF-α (Fig 5E), and IL-12 (Fig 5F) in the lung tissue.

## Discussion

We developed peptides derived from the CTLD4 domain of the human mannose receptor, MRC-1, that interacts with the conserved cholesterol-binding loop of CDCs, which is critical for toxin binding to eukaryotic cells. Out of the eight peptides that we screened, two peptides showed the highest binding to purified CDCs and protected host cells against toxin-induced cytolysis and inflammation. We show that the peptides were effective against the three major bacterial CDCs, PLY, LLO, and SLO. The peptides also bound *in situ* to PLY-producing pneumococci and blocked hemolysis, suggesting that the peptides target both secreted and pneumococcal-localized PLY, since PLY has also been shown to be exposed on bacterial surfaces (Price & Camilli, 2009; Shak *et al*, 2013). Since CDCs are one of the major conserved bacterial virulence factors, targeted neutralization of their effects is an innovative approach to eliminate the bacteria without killing them, and hence, the risk for resistance development

is lower (Escajadillo & Nizet, 2018). Previously, antibodies targeting PLY, cholesterol-loaded decoy liposomes, and phytosterols resembling cholesterol, have been found to protect mice against *S. pneumoniae* infection (Musher *et al*, 2001; Henry *et al*, 2015; Li *et al*, 2015). However, unlike the MRC-1 peptides, the above strategies can cause undesired immune activation since they are not host-derived.

Besides inducing cytolysis, CDCs have been shown to promote bacterial invasion and entry into host cells in a dynamin-dependent and F-actin-dependent manner (Vadia *et al*, 2011). We found that our MRC-1 peptide effectively blocked bacterial invasion in the 3D lung epithelial model, as well as in primary human DCs. The ability of the peptide to reduce bacterial invasion was similar as blockage of PLY using antibodies. The human macrophage and DC receptor MRC-1 has been shown to promote bacterial uptake into phagosomes without culminating in lysosomal fusion, thereby providing a safe intracellular niche for bacteria (Sweet *et al*, 2010; Rajaram *et al*, 2017; Subramanian *et al*, 2019b). Since the peptides are derived from the CTLD4 domain of MRC-1 that interacts with the CDCs, we hypothesized that they might block bacterial uptake mediated by PLY-MRC-1 interaction. In DCs treated with the MRC-1 peptide, intracellular pneumococci did not co-localize with MRC-1, but co-stained with the autophagy marker LC3B, indicating that the bacteria are targeted to autophagy killing. Many bacterial pathogens avoid immune clearance by persisting intracellularly within host cells and contribute to the relapse of the infection (Dusane *et al*, 2018; Ercoli *et al*, 2018). Hence, the peptides could be used to eliminate bacteria that have escaped antibiotic killing by persisting intracellularly within MRC-1-positive tissue macrophages and DCs.

Since peptides are prone to enzymatic degradation *in vivo*, we developed CaP NP-loaded with the MRC-1 peptide that allowed rapid and efficient targeting to the lungs after intranasal administration. CaP NPs are non-toxic and have been successfully used to deliver bioactive molecules like peptides (Miragoli *et al*, 2018) and microRNA (Di Mauro *et al*, 2016) owing to their cellular permeability. However, the pharmacokinetics properties and the therapeutic window of the peptide nanocarriers need to be investigated for clinical application of the peptides. Further, any possible effects due to interaction of NPs with bacteria are not completely ruled out and need further studies.

In conclusion, our data reveal that MRC-1-derived peptides bind to bacterial pore-forming toxins, inhibit lysis of host macrophages, and reduce inflammation. They also block MRC-1-mediated bacterial uptake into DCs and promote autophagy killing of intracellular bacteria. Using two *in vivo models*, zebrafish and mice, we found that administration of peptide-conjugated NPs enhanced survival

against pneumococcal infection as well as reduced the bacterial load and inflammation in the lungs. We envisage that these toxin-binding peptides could be used in combination with antibiotics to treat patients with bacterial infections to neutralize the cytotoxicity and inflammation induced by pore-forming toxins. During the 1918 Spanish flu pandemic, severely ill influenza A virus-infected patients often developed secondary pneumonia caused by bacterial respiratory pathogens, primarily *S. pneumoniae* and *S. pyogenes*. Therefore, we also suggest that intranasal delivery of peptide-coated NPs might be particularly useful in cases with acute respiratory distress syndrome where secondary pneumonia is suspected. However, this remains to be investigated in future studies.

# Materials and Methods

### Study design

The main objective was to construct peptides from the human mannose receptor, MRC-1, as decoys to inhibit bacterial CDCs and treat pneumococcal infection. The study design consisted of (i) computational modeling of interaction between MRC-1 and bacterial CDCs and designing peptides from region of interaction, (ii) *in vitro* cell culture and 3D lung tissue models to screen peptide efficacy, (iii) synthesis of peptide-conjugated nanoparticles for intranasal delivery of peptides, and (iv) *in vivo* validation of therapeutic peptide nanocarriers using zebrafish and mice infection models. First, we used the zebrafish (*Danio renio*) embryo model to assess the protection conferred by peptides against pneumococcal infection. To rule out non-specific effects of the peptide, a control peptide which does not bind the bacterial toxin was included in the zebrafish study. The zebrafish embryos were randomly assigned to the treatment groups and a minimum of 156 embryos/group were used. All experiments were performed thrice. The zebrafish study was approved by the ethical review board, Stockholm Animal Research Committee (Drn 19204-2017) and the Swedish Board of Agriculture. Next, we used the mouse model to validate the efficacy of peptide nanocarriers using an intranasal infection model. Six-week-old male C57BL/6 wild-type mice were used (Charles River, Germany). The mouse experiments were randomized and included 10 mice per treatment group. The sample size was determined based on previous experience (Subramanian *et al*, 2019b) and accounting the mortality rate. Appropriate controls were included to exclude unspecific effects of the peptide or nanoparticles alone. The mouse experiments were approved by the local ethical committee (Stockholms Norra djurförsöksetiska nämnd).

### Bacterial strains and growth conditions

The encapsulated *S. pneumoniae* serotype 4 strain TIGR4 (T4; ATCC BAA-334) and the type 2 strain D39 were used in this study, as well as the isogenic capsule and pneumolysin deletion mutants, T4R (Fernebro *et al*, 2004), T4Δ*ply*, and T4RΔ*ply* (Littmann *et al*, 2009), respectively. Pneumococci were grown on blood agar plates overnight and colonies were inoculated into pre-warmed C + Y medium and grown to OD = 0.4 at 37°C. The *Streptococcus pyogenes* strain 5,548 of serotype M1T1 and the isogenic SLO mutant were obtained from Prof. Victor Nizet, University of California, San Diego.

*Streptococcus pyogenes* was grown on brain heart infusion agar plates overnight and inoculated into Todd Hewitt Broth containing 0.5% yeast extract at 37°C. *Listeria monocytogenes* type 1 strain (ATCC 19111) was obtained from the Swedish Institute for Infectious Disease Control and grown on brain heart infusion agar plates overnight and inoculated into on brain heart infusion broth at 37°C.

### Purified bacterial toxins and MRC-1 peptides

Recombinant toxins, PLY, LLO, and SLO, were expressed and purified at the Protein Production Platform, Nanyang Technological University, Singapore. The purified endotoxin-free toxoid derivative, Pdb PLY (W433F), was a gift from Prof. Aras Kadioglu, University of Liverpool, UK. The purified LLO toxoid derivative, LLO (W489F), was a gift from Prof. Gregor Anderluh, National Institute of Chemistry, Slovenia. The overlapping 13-mer MRC-1 peptides (described in Appendix Table S1) were commercially synthesized and purified to > 95% purity by Genscript (NJ, USA). The lyophilized peptides were dissolved in ultrapure water or analytical grade dimethyl sulfoxide (Sigma) according to the manufacturer's suggestions.

### Mouse infection model

All animal experiments were approved by the local ethical committee (Stockholms Norra djurförsöksetiska nämnd). Six-week-old male C57BL/6 wild-type mice were used (Charles River, Germany). The study included 10 mice/individual group which were randomly assigned to the different treatment groups. Sample size calculations were determined according to previous experience (Subramanian *et al*, 2019b). Mice were anesthetized by inhalation of 4% isofluorane (Abbott) mixed with oxygen and intranasally administered with 50 μl of PBS containing $2 \times 10^6$ CFU of *S. pneumoniae* T4, or the PLY mutant T4Δ*ply*, or T4 mixed with the peptide P2 alone (5 μg/mouse), or P2 conjugated to CaP NPs (5 μg peptide; 25 μg CaP NPs/mouse) or blank NPs (25 μg/mouse). Clinical symptoms of mice were monitored for 3 days post-infection and sacrificed upon reaching humane end points according to ethical regulations. The mice were checked at least twice every day; morning at 9 am (referred as "early check") and evening at 7 pm (referred as "late check"). Post-mortem, lungs were collected and washed in PBS and homogenized by passing through the 100-μm cell strainer. Bacterial counts were determined from lung homogenates by viable count on blood agar plates. Aliquots of lung homogenates were frozen at −80°C for cytokine quantification by ELISA.

### IVIS imaging to visualize biodistribution of peptide-conjugated NPs in mice

To verify the delivery of peptide-conjugated NPs to the lungs, mice were anesthetized by inhalation of 4% isofluorane mixed with oxygen and administered intranasally (50 μl/mouse) with Cy7-conjugated peptide P2-NPs (5 μg peptide; 25 μg CaP NPs/mouse) or non-fluorescent NPs alone (negative control), or Cy7 dye alone (positive control) or PBS. Mice were sacrificed either at 1 h or 24 h post-treatment. Lungs were collected post-mortem. Cy7 fluorescence in the mouse lungs was imaged using the IVIS Spectrum-CT Imaging system (Caliper-Perkin Elmer). RGB Profile Plot representing the signal intensity was generated for each image taken with the IVIS Spectrum System (Sierakowiak *et al*, 2019).

## Zebrafish infection model

The zebrafish study was approved by the ethical review board, Stockholm Animal Research Committee (Drn 19204-2017) and the Swedish Board of Agriculture. The AB strain of zebrafish embryos (*Danio renio*) was collected within first hours post-fertilization (hpf) from the zebrafish core facility at Karolinska Institutet, Stockholm, and maintained at 28.5°C in E3 medium (5.0 mM NaCl, 0.17 mM KCl, 0.33 mM CaCl$_2$, 0.33 mM MgSO$_4$). The fertilized embryos (3–4 hpf) were microinjected into the yolk sac with 1 nl of E3 medium containing 500 CFU of *S. pneumoniae* T4 alone or T4 mixed with peptide P2 or control peptide CP2 (1 nM) or T4 mixed with P2-conjugated NPs or blank NPs. Microinjection was done using a glass needle (Harvard apparatus, Quebec, Canada) controlled with a micromanipulator Narishige MN-153 (Narishige International Limited, London, UK) connected to an Eppendorf FemtoJet express (Eppendorf AG, Hamburg, Germany). The injected volume was previously optimized by injecting of a droplet into mineral oil over a scale bar. The embryos were randomly assigned to the treatment groups and a minimum of 156 embryos/group was used. To determine the actual number of bacteria in the injected volume, one drop was collected into the agar plates, and 1–3 embryos were digested and plated just immediately after injection. The infected embryos were incubated at 30°C for 96 h. The infected embryos were monitored for disease symptoms and survival under a stereomicroscope twice a day up to 4 days post-injection. Absence of heartbeat and movement was interpreted as a sign of death. All experiments were performed thrice.

## Primary monocyte-derived dendritic cells, cell culture, and infection

Human DCs were differentiated from primary monocytes isolated from anonymous buffy coats of healthy blood donors (Karolinska University Hospital). The monocytes were isolated by using the RosetteSep monocyte purification kit (Stem Cell Technologies) as described previously (Subramanian *et al*, 2017). For differentiation into DCs, monocytes were cultured in R10 (RPMI 1640, 2 mM L-glutamine, 10% FBS) supplemented with GM-CSF (40 ng/ml) and IL-4 (40 ng/ml) from Peprotech for 6 days. DCs were verified by flow cytometry to be > 90% CD1a$^+$ CD11c$^+$. For infection, DCs were incubated with bacteria at MOI of 10 and the extracellular bacteria were killed using gentamicin (200 µg/ml) after 2 h post-infection.

Human monocytic leukemia THP-1 cells (ATCC TIB-202) were cultivated in R10 medium and maintained at density of 10$^6$ cells/ml. For differentiation into macrophages, THP-1 cells were treated for 48 h with 20 ng/ml of phorbol myristate acetate (Sigma). For cytokine measurements, differentiated THP-1 macrophages were incubated with purified PLY, LLO, and SLO (0.5 µg/ml) in the presence of 100 µM peptides P2, P3, and control peptide, CP2, and the culture supernatants were collected 18 h later for cytokine ELISA.

## 3D lung epithelial tissue model

The organotypic lung epithelial tissue model was set up as previously described (Nguyen Hoang *et al*, 2012). The human lung fibroblast cell line, MRC-5 (ATCC CCL-171) was cultured to between 80 and 90% confluence and maintained in Dulbecco's modified Eagle's medium (DMEM) (GE Healthcare Life Sciences, Marlborough,

MA) supplemented with 10% heat-inactivated fetal bovine serum (FBS) (Sigma Aldrich, St. Louis, MO), 2 mM L-glutamine (GE), 1 mM sodium pyruvate (GE), and 10 mM HEPES buffer (GE). The human lung epithelial cell line 16HBE14o- (gift from Dr. Dieter Gruenert, Mt. Zion Cancer Center, University of California, San Francisco, CA) was modified to express GFP (Nguyen Hoang *et al*, 2014) and cultured in a tissue-culture flask coated with fibronectin solution containing 1 mg/ml bovine serum albumin (0.1%), 3 mg/ml bovine collagen type I (Advanced BioMatrix, San Diego, CA), and 1 mg/ml human fibronectin (Corning Inc., Corning, NY) in PBS. GFP-expressing 16HBE14o- (henceforth, GFP-16HBE) was grown to between 80 and 90% confluence in Minimum Essential Medium (MEM) (Thermo Fisher Scientific, Waltham, MA, Gibco) containing 10% FBS, 10 mM HEPES buffer, 2 mM L-glutamine, and 1× nonessential amino acids (Gibco).

The model was prepared by first seeding an acellular collagen layer (1 ml) in 6-well plate Transwell inserts (Corning Inc.) and then allowing the layer to gel for 30 min at 37°C and 5% CO$_2$. After gelation, a cellular layer (3 ml) containing MRC-5 cells suspended in a collagen matrix was added to the model and allowed to gel for 2 h prior to the addition of complete DMEM. Media was changed the next day and subsequently every second day for 6–7 days until the MRC-5 cells were fully embedded in the collagen matrix. Once ready, the apical media was removed and 50 µl of GFP-16HBE cells were added to the apical side of the models at a density of $1.6 \times 10^6$ cells/ml (80,000 cells/model), allowed to settle and adhere for 2 h, and then, the complete model was submerged in complete DMEM for 3–4 days until confluent. Once confluent, models were airlifted and maintained at an air–liquid interface (ALI) for a minimum of 5 days prior to stimulation. The acellular collagen layer solution was composed of complete DMEM, 3 mg/ml bovine collagen type I, and a premix solution containing 5× DMEM, 2 mM L-glutamine, 71.2 mg/ml NaHCO$_3$, 45% FCS, and 50 mg/ml gentamicin (Sigma Aldrich). The cellular layer contained the premix solution, 3 mg/ml bovine collagen type I, complete DMEM, and MRC-5 cells at a density of $2.3 \times 10^5$ cells/ml (75,000 cells/model). Models were maintained in complete DMEM for the entirety of the experiment.

## Hemolysis assay

Human blood from anonymous healthy donors (obtained from Karolinska University Hospital) was diluted 1:100 in PBS with 0.5 mM DTT and mixed 1:1 with twofold serial dilutions of 10$^8$ CFUs of bacteria or 1 µg/ml purified bacterial toxins in 96-well plates. The MRC-1 peptides were serially diluted 10-fold (1–1,000 µM) in PBS and added to the wells prior to addition of blood. The blood was co-incubated with whole bacteria or purified toxins at 37°C for 1 h, and after 50 min, 0.1% triton X-100 was added to the positive control wells. Cells were spun down at 400 *g* for 15 min and the absorbance of the supernatants was measured at 540 nm in a microplate reader. Percentage of lysis compared to the toxin alone was calculated. All samples were assayed in triplicates.

## ELISA to test binding of peptides to PLY, LLO, and SLO

Briefly, 96-well flat-bottomed plates (Sigma, UK) were coated overnight with 10 µM of MRC-1 peptides in coating buffer (15 mM Na$_2$CO$_3$, 35 mM NaHCO$_3$, pH 9.6). Wells were blocked with 200 µl

of 10% (v/v) FBS in PBS for 2 h and then washed three times with PBS, 0.05% (v/v) Tween 20 (Sigma). 50 μl of purified PLY, LLO, and SLO (0–1 μM) in PBS was added and incubated at 37°C for 1 h. 1 μM BSA was added to control wells. Wells were washed with PBS and bound proteins were detected by adding 100 μl of mouse α-PLY (1:1,000), mouse α-SLO (1:400), and rabbit α-LLO (1:1,500) (Abcam), respectively, in blocking buffer. Plates were incubated with 100 μl of secondary anti-rabbit/anti-mouse IgG HRP (1:2,000) dilution in blocking buffer. Bound antibodies were detected by adding 100 μl of tetramethylbenzidine substrate solution for 30 min. Phosphoric acid (1 M) was used as the stop solution and absorbance was measured at 450 nm.

### LDH cytotoxicity assay

Cytotoxicity was determined by measuring the activity of lactate dehydrogenase enzyme released into the culture supernatant using the Cytotoxicity Detection kit (Roche) according to the manufacturer's instructions. The percentage cytotoxicity was calculated by ratio to 100% lysis control (0.1% saponin).

### Cytokine ELISA

For cytokine assays, the cell-free culture supernatants were collected 18 h post-stimulation and frozen at −20°C. The levels of TNF-α, IL-8, and IL-12 (p70) in the culture supernatants and mouse lung tissue were analyzed using the OptEIA ELISA kit (BD Biosciences) following the manufacturer's instructions.

### Co-localization of MRC-1 with purified PLY, LLO, and SLO in DCs

For co-localization studies with purified toxins, human DCs (3 × 10^5) were attached on poly-lysine-coated coverslips and incubated with 0.2 μg/ml of purified PLY, LLO, and SLO for 45 min at 37°C and washed twice with PBS. The toxoid derivatives, PLY (W433F) and LLO(W489F), were used as negative controls. The cells were fixed with PBS-buffered 4% paraformaldehyde for 10 min. The cells were permeabilized with 0.5 for 15 min in dark. Non-specific interactions were blocked by incubating cells with 5% FBS in PBS for 30 min. The cells were then incubated overnight with Alexa 647-conjugated rabbit anti-EEA-1 (Abcam; 1:50 dilution) to stain early endosomes. PLY and SLO was detected using mouse anti-PLY (Abcam; 1:100 dilution) or mouse anti-SLO (Abcam; 1:100 dilution) followed by secondary goat anti-mouse secondary antibody (Thermo Fisher Scientific; 1:500 dilution). LLO was detected using rabbit anti-LLO conjugated to Alexa 488 using the Zenon Rabbit IgG Labeling kit (Abcam; 1:100 dilution). MRC-1 was detected using Alexa 594-conjugated Rabbit anti-MRC1 (Abcam; 1:100 dilution). The coverslips were mounted on slides using Prolong Gold anti-fade mounting medium containing the nuclear stain 4,6-diamidino-2-phenylindole (DAPI; Thermo Fisher Scientific). Images were acquired using a Delta Vision Elite microscope under the 100× oil immersion objective (GE Healthcare).

### Computational docking of MRC-1 CTLD4 with bacterial toxins

Computational docking of the CTLD4 domain of MRC-1 with the pore-forming toxins PLY, LLO, and SLO was performed using the

## The paper explained

### Problem

Cholesterol-dependent cytolysins (CDCs) are bacterial toxins that bind to cholesterol on target cell membrane and form pores resulting in cell lysis and inflammation. CDCs are major virulence factors for several bacteria such as the human respiratory pathogen, *S. pneumoniae*. The structures of CDCs are conserved and hence represent attractive targets for novel treatments that neutralize toxin function. We recently identified that the human mannose receptor, MRC-1, binds to pneumococcal toxin, pneumolysin. In this study, we developed peptides from the region of interaction between MRC-1 and CDCs to inhibit toxin activity and treat pneumococcal disease.

### Results

Using computational docking, we designed peptides from the CTLD4 domain of MRC-1 that bind to the cholesterol-binding loop of the major CDCs: PLY, LLO, and SLO. We screened peptides for toxin inhibition and identified peptides that dose-dependently bind and inhibit toxin-induced cytolysis and inflammation. Also, the peptides inhibit bacterial internalization into DCs via MRC-1 and target intracellular bacteria to autophagy killing. Further, we developed biocompatible calcium phosphate nanoparticles conjugated to peptides for enhanced bioactivity and intranasal delivery. We found that the peptide-conjugated nanoparticles enhanced survival and reduced the bacterial burden and inflammation in mouse and zebrafish models of pneumococcal infection.

### Impact

Our study suggests that the nanoparticles conjugated to MRC-1 peptides could be used as an adjunctive therapy together with antibiotics to treat bacterial infections.

ClusPro 2.0 docking server (Kozakov et al, 2017). The available crystal structures of MRC-1 CTLD4 (PDB id-1EGG), PLY (PDB id-5CR6), LLO (PDB id-4CDB), and SLO (PDB id-4HSC) were obtained from the Protein Data Bank. Docking was performed using the balanced model, considering electrostatic and hydrophobic interactions. The models were ranked based on the size of the cluster that is defined as the ligand position with the most neighbors within 9 Å distance. The docking models were analyzed using the PyMOL Molecular graphics software version 2.0.6. The receptor (MRC-1) was colored red, and the ligands PLY, LLO, and SLO were colored green. The eukaryotic membrane-binding undecapeptide loop in domain 4 of PLY, LLO, and SLO was labeled yellow. The zoomed structures show the amino acid residues in MRC-1 (red) and the undecapeptide loop of PLY, LLO, and SLO (yellow) that are predicted to involve in hydrogen bonding interactions with each other.

### CFU plating assay to quantify intracellular bacteria in DCs

Briefly, DCs were infected with pneumococci, type 2 (D39) or type 4 (T4) bacteria at MOI of 10 with or without adding 100 μM of the MRC-1 peptides, P2 or CP2. At 2 h post-infection, gentamicin (200 μg/ml) was added and incubated for 1 h at 37°C to kill extracellular bacteria. The anti-PLY antibody (1:100) was used as control to ascertain the role of PLY in bacterial invasion into DCs. The actin polymerization inhibitor, cytochalasin D (0.5 mM), was used as control to inhibit bacterial uptake by DCs. The cells were washed,

resuspended in 100 µl PBS, and serial dilutions were plated on blood agar plates in 10 µl volume and incubated overnight.

### Statistical analysis

For *in vivo* studies, the sample size of 10 mice/group was chosen based on our previous experience with pneumococcal intranasal infection model. For zebrafish experiments, a minimum sample size of 156 embryos/group was used. Mice were age and sex matched to minimize any bias between treatment groups. No blinding was done, but the mice were randomly assigned to the treatment groups. To avoid bias, the infection was done by one author and downstream analysis by another. The zebrafish experiments were performed in another laboratory. Data were statistically analyzed using GraphPad Prism v.5.04. Data represent mean ± s.e.m. For comparison between groups, the one-way or two-way ANOVA with Bonferroni or Dunnett's *post hoc* test for multiple comparisons was used. Comparison of survival curves in mice and zebrafish was performed using the Log-rank (Mantel–Cox) test. Normalized data were analyzed using paired *t*-tests. Differences were considered significant at $*P < 0.05$, $**P < 0.01$; $***P < 0.001$, $****P < 0.0001$. n.s. denotes that the difference is not significant. The exact *P* values are reported in Appendix Table S4.

## Data availability

All data reported in the paper are included in the manuscript or available in the Appendix. Other data that support the findings of this study are available from the corresponding authors upon request.

**Expanded View** for this article is available online.

### Acknowledgements

This work was supported by grants from the Swedish Research Council, Stockholm County Council, the Swedish Foundation for Strategic Research (SSF), and the Knut and Alice Wallenberg foundation (to B.H.N). S.B.B. was supported by the National Science Foundation Graduate Research Fellowship Program under Grant No. DGE-1256082. Funding from the European Research Council (ERC) under the European Union's Horizon 2020 research and innovation program (ERC Grant agreement no. 758705) and the Torsten Söderberg Foundation (M87/18) is kindly acknowledged. We thank Prof. Aras Kadioglu, University of Liverpool, and Prof. Gregor Anderluh, National Institute of Chemistry, Slovenia, for kindly providing us the mutant toxoid derivatives of PLY and LLO, respectively. We also thank Prof. Victor Nizet, University of California, San Diego, for providing the *Streptococcus pyogenes* strain 5,548 and its isogenic SLO mutant. We thank the Histological core facility at Karolinska Institutet for help with lung histology. We greatly acknowledge the Protein Production Platform, at Nanyang Technological University, Singapore, for expression and purification of bacterial toxins.

### Author contributions

KS, PB, GAS, MS, and BH-N designed the study. KS, FI, VT, PM, SA, M-SA, and SBB. performed the experiments. KS and BH-N wrote the manuscript. All authors contributed to writing and have approved the final version of the manuscript.

### Conflict of interest

The authors declare that they have no conflict of interest.

### For more information

American lung association: https://www.lung.org/lung-health-diseases/lung-disease-lookup/pneumonia

Karthik Subramanian scholar profile: https://scholar.google.se/citations?user=4Ez50KwAAAAJ&hl=en&oi=ao#

The laboratory of Birgitta Henriques-Normark: https://ki.se/en/mtc/birgitta-henriques-normark-group

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
