## [Review Process File · EMBO Molecular Medicine]

Mannose receptor-derived peptides neutralize pore-forming toxins and reduce inflammation and development of pneumococcal disease

Karthik Subramanian, Federico Iovino, Vasiliki Tsikourkitoudi, Padyrk Merkl, Sultan Ahmed, Samuel Berry, Marie-Stephanie Achtgen, Mattias Svensson, Peter Bergman, Georgios Sotiriou, and Birgitta Henriques-Normark

DOI: [10.15252/emmm.202012695](https://doi.org/10.15252/emmm.202012695)

Corresponding authors: Birgitta Henriques-Normark (birgitta.henriques@ki.se)

Review Timeline:

Submission Date:	7th May 20
Editorial Decision:	5th Jun 20
Revision Received:	30th Jul 20
Editorial Decision:	26th Aug 20
Revision Received:	30th Aug 20
Accepted:	2nd Sep 20

Editor: Zeljko Durdevic

Transaction Report:

5th Jun 2020

Dear Prof. Henriques-Normark,

Thank you for the submission of your manuscript to EMBO Molecular Medicine. We have now heard back from the three referees who agreed to evaluate your manuscript. As you will see from the reports below, the referees acknowledge the interest of the study. However, they raise some concerns that should be addressed in a major revision of the present manuscript. Addressing the reviewers' concerns in full will be necessary for further considering the manuscript in our journal.

Acceptance of the manuscript will entail a second round of review. Please note that EMBO Molecular Medicine encourages a single round of revision only and therefore, acceptance or rejection of the manuscript will depend on the completeness of your responses included in the next, final version of the manuscript. For this reason, and to save you from any frustrations in the end, I would strongly advise against returning an incomplete revision.

We realize that the current situation is exceptional on the account of the COVID-19/SARS-CoV-2 pandemic. Therefore, please let us know if you need more than three months to revise the manuscript.

I look forward to receiving your revised manuscript.

***** Reviewer's comments *****

Referee #1 (Remarks for Author):

This manuscript describes the structural details of the interaction of cholesterol dependent cytotoxins, CDCs, with the mannose receptor C-1. The authors published a manuscript recently that describes the discovery of this interaction which is unique as it is protein and not carbohydrate dependent. The series of studies in that paper closely follow the layout of the current paper and confirm the conclusions thereof. The structural biology of the interaction is examined in more detail in the current work while the events downstream of MRC-1 binding in leukocytes were more the focus of the first paper.

The authors present a very comprehensive series of experiments, focusing largely on pneumolysin, to show the regions of the toxin and the receptor that interact, and the inhibition of the major cytolytic functions of the CDCs by peptides derived from the MRC-1 binding site. The studies are

very well controlled and many of the experiments are verified by several different methods. They use this information to then design a set of peptides that can competitively inhibit the binding and thus disrupt cytotoxicity as a therapeutic option. These effects are shown in vitro and in zebrafish and mice.

The large amount of cross-confirmatory data support all the conclusions. The findings are clearly novel and the therapeutic angle is potentially a new approach to aspects of disease attributable to the toxin during disease.

Comments:

- 1) The introduction could be expanded to describe MRC-1 function. While this is described in the first paper, a short version would also be helpful in this paper.
- 2) Does pneumolysin bind to and lyse cells lacking MRC-1? This would address the degree to which this interaction has functional consequences distinct from those arising by the classical CDC's direct interaction with cholesterol.
- 3) Pneumolysin is a major virulence determinant but there is no animal model that specifically focuses on this toxicity. In contrast, necrotizing fasciitis and botulism are examples of singularly toxin mediated infection where outcome could be hugely impacted by the nanotherapy proposed in the paper. It would be dramatic and therapeutically important to show efficacy of the therapy in at least one of these models.
- 4) The natural ligands for MRC-1 are multiple including sulfated glycoconjugates, complex saccharides, and collagen. Is there interference with toxin binding or changes in physiological outcomes when these other sites are occupied?
- 5) Fig 3E shows a very low resolution study of the ability of the toxin to be neutralized by the peptide. A higher resolution study focusing on the dose response to inhibit hemolysis by pure toxin vs peptide would be important
- 6) , specifically to show potency which is measured by the dose where the hemolysis begins to be affected, the slope of the inhibition to determine the MD50 and the plateau at full effect to show neutralization can be 100%.

Referee #2 (Remarks for Author):

This interesting study by Subramanian et al. builds on the previous identification of MRC-1 as an immune cell receptor for pneumolysin (Ply). The authors identify peptides derived from MRC-1 that bind to Ply, as well as to other related toxins such as LLO and SLO. These peptides diminish the activities of Ply, such as cytolysis, proinflammatory cytokine production, epithelial integrity and MRC-1-mediated internalization. These peptides, like genetic ablation of ply, resulted in targeting of bacteria to autophagosomes and enhanced bacterial killing. The linkage of a peptide to nanoparticles resulted in a reagent that increased bacterial clearance and host survival in zebrafish and mouse models of infection, albeit with relatively modest changes in disease outcome.

The authors should address the following:

1. Specificity of inhibitory peptides. (a) Fig. S2C shows that all MRC-1 peptides appear to have some inhibitory effect on hemolysis. How do the authors explain the inhibitory effect on MRC-1 peptides that do not interact with the loop? Note that the peptides each have more acidic than basic residues; one of the two control peptides shares this property. A superior control peptide would be a 'scrambled' version of P2 or P3, depending on which peptide is being tested. Such a peptide need not be tested in all of the assays described, but a few key assays that demonstrated that the

- peptide sequence rather than just its amino acid composition. (c) The authors choose to focus on two, i.e. P2 and P3. A statistical analysis of P2 and P3 vs. other peptides should be performed. (d) Fig. 3D. What explains the surprising punctate staining pattern of Sp by P2? Is it dependent on capsule, i.e. does the acapsular mutant give the same pattern? Is the same staining pattern also revealed using anti-Ply antibody?
2. P. 4, line 7. Does co-localization of LLO and SLO with EEA-1 require MRC-1, as they have shown previously for PLY utilizing knock-down of MRC-1?
 3. Explanation of Ply concentrations utilized, a particularly important issue that this group addressed in a previous publication. (a) P. 7, 5th line from bottom. Concentration of Ply utilized was 500 ug/ml, or 2.5 x the concentration typically utilized in previous publication. Can the authors provide an explanation of how concentration was chosen, given the concentration-dependent response to CDC? (b) P. 9. middle of page. What is final concentration when 50 ug are added to tissue model? The authors should describe rationale for choosing this concentration. (c) Fig. 3C and 4A. The concentration of Ply used when measuring cytokine production causes 90% cytotoxicity. How might this alter the production of cytokines?
 4. Fig. 4D and E. (a) Can the authors rule out the possibility that it is bacterial intracellular survival that is diminished by P2 or by the Ply-deficient strain? (b) It is not clear how efficient bacterial invasion is. The authors should convert CFU/ml into % entry to convey this information.
 5. P. 11, bottom and Fig. 4F and G. How do the authors reconcile their findings with those of Omishi and coworkers (Cell Reports '20 and Autophagy '20) and others who report that Ply triggers autophagy in a variety of cell types?
 6. The panels comparing T4R vs. T4R delta ply are difficult to compare because color of LC3B is different in the two panels.
 7. Fig. 5. Can the authors rule out that the greater efficacy of P2-NPs compared to NP's or P2 alone is due to the agglutination of bacteria by the beads rather than specific inhibition of PLY activity?
Minor points:
 8. P. 6, last line. the authors should describe how they determined which residues interact with the loop 4. (It is clear that they used modeling, but what were the criteria for assigning interacting residues?)
 9. P. 6. line 7. The "cholesterol binding loop" alters the avidity of cholesterol binding by altering oligomerization, rather than directly binding cholesterol (see Dowd and Tweten, PLoS Pathogen, 2012); line 12 and elsewhere should also be edited accordingly.
 10. P. 6, line 4. "show" should read "suggest"
 11. P. 6, line 6. "we found that tryptophan" should read "modeling predicted that tryptophan"
 12. Fig. 5B and C are difficult to discern the different groups.

Referee #3 (Remarks for Author):

Very elegant work showing that peptide binding/masking of the pneumococcal toxin pneumolysin influences the intracellular fate of the bacteria. Blocking of the ply dependent inhibition of phagosome maturation led to successful autophagy of pneumococci, which in the presence of active ply could survive in the cells.

Main comments:

Figure 1 shows very convincingly colocalization of PLY and LLO with MRC-1 and EEA-1, but SLO appears to colocalise with MRC-1, but not EEA-1. Showing a bar chart showing the extent of colocalization would solve the issue and allow to clearly report the results in the text. When comparing panels in Figure 4F to those of 4G, it would appear that there is much more red

stain (mannose receptor) in the left panels and much more pink stain in the right panels, as if the interaction with ply could somehow have upregulated mannose receptor and/or downregulated LC3B. As such an effect could be caused by indirect effects or cause downstream indirect effects, it would be important if authors could quantify the reactivity in the images/samples and confirm that the markers are equally expressed or distributed.

Minor comments:

The scale ticks and labelling of the X-axis of Fig 3E appear to be wrong. onse?

Response to Reviewers- EMM-2020-12695

Reviewer #1 (Remarks to the Author)

This manuscript describes the structural details of the interaction of cholesterol dependent cytotoxins, CDCs, with the mannose receptor C-1. The authors published a manuscript recently that describes the discovery of this interaction which is unique as it is protein and not carbohydrate dependent. The series of studies in that paper closely follow the layout of the current paper and confirm the conclusions thereof. The structural biology of the interaction is examined in more detail in the current work while the events downstream of MRC-1 binding in leukocytes were more the focus of the first paper. The authors present a very comprehensive series of experiments, focusing largely on pneumolysin, to show the regions of the toxin and the receptor that interact, and the inhibition of the major cytolytic functions of the CDCs by peptides derived from the MRC-1 binding site. The studies are very well controlled and many of the experiments are verified by several different methods. They use this information to then design a set of peptides that can competitively inhibit the binding and thus disrupt cytotoxicity as a therapeutic option. These effects are shown in vitro and in zebrafish and mice. The large amount of cross-confirmatory data support all the conclusions. The findings are clearly novel and the therapeutic angle is potentially a new approach to aspects of disease attributable to the toxin during disease.

Author Response: We thank the reviewer for the positive comments!

1) The introduction could be expanded to describe MRC-1 function. While this is described in the first paper, a short version would also be helpful in this paper.

Author Response: Thank you for this suggestion. Accordingly, we have now added a short description on MRC-1 function in the revised manuscript text (page 4, lines 7-13).

2) Does pneumolysin bind to and lyse cells lacking MRC-1? This would address the degree to which this interaction has functional consequences distinct from those arising by the classical CDC's direct interaction with cholesterol.

Author Response: Thank you for this relevant question. Yes, pneumolysin (PLY) is a cholesterol-binding toxin and can virtually bind to all human cells that express membrane cholesterol, independently of MRC-1 expression. While interaction with cholesterol is involved during pore-formation, PLY at sublytic doses, induces a cell-death independent anti-inflammatory response specifically in cells that express MRC-1, e.g. dendritic cells, lung alveolar macrophages and BMDMs. This immunomodulatory effect was absent in cells lacking MRC-1 deficient cells and knockout mice (Subramanian *et al*, 2019). Hence, the interaction of PLY with MRC-1 has a functionally different consequence from the classical cholesterol interaction. In this study, we investigated the inhibitory effect of peptides against PLY-induced cytolysis and inflammation in many cell types, irrespective of MRC-1 expression. Hence, we have used both MRC-1 positive (human dendritic cells) as well as negative cells (THP-1 macrophages, lung epithelial cells, whole blood) to test the efficacy of the peptides in inhibiting PLY-induced cytolysis.

3) Pneumolysin is a major virulence determinant but there is no animal model that specifically

focuses on this toxicity. In contrast, necrotizing fasciitis and botulism are examples of singularly toxin mediated infection where outcome could be hugely impacted by the nanotherapy proposed in the paper. It would be dramatic and therapeutically important to show efficacy of the therapy in at least one of these models.

Author Response: Thank you for the suggestion, but we would like to clarify that animal models for pneumolysin toxicity are well established. Many studies have previously demonstrated the importance of PLY toxicity in pneumococcal disease development using mutant strains in murine models (Garcia-Suarez Mdel *et al*, 2004) (Garcia-Suarez Mdel *et al*, 2007) (Witzenrath *et al*, 2006). Besides, in the current study (Figs. 5A, C), we have compared wild-type and PLY mutant strains and found reduced virulence of the PLY mutant *in vivo* using mice and zebrafish models. Also, we have demonstrated the effectiveness of the peptides against Streptolysin O produced by *Streptococcus pyogenes in vitro*. However, the models for necrotizing fasciitis would be tricky to use since the pathology also involves a non-CDC peptide toxin Streptolysin S (Humar *et al*, 2002). Similarly, the botulinum toxin is not a CDC toxin, but rather a neurotoxin that binds to presynaptic nerve terminals and has a completely different mechanism of action. Anyway, thanks for the suggestion, even though we think that the PLY model is most relevant here.

4) The natural ligands for MRC-1 are multiple including sulfated glycoconjugates, complex saccharides, and collagen. Is there interference with toxin binding or changes in physiological outcomes when these other sites are occupied?

Author Response: This is an interesting question. MRC-1 consists of multiple domains that interact with a variety of ligands such as sulphated glycoconjugates, saccharides and collagen (Fig. EV2a). The N-terminal cysteine rich domain binds to sulphated glycoconjugates while the fibronectin type II domain binds collagen. However, we found that specifically domain 4 of pneumolysin binds to C-type lectin domains 4-7 on MRC-1. In our previous study, using competition ELISA, we showed that MRC-1 still binds to PLY even in the presence of capsular polysaccharides (Figure 1 below adapted from (Subramanian *et al.*, 2019). Further, we also showed that both wild-type encapsulated pneumococci as well as a capsular mutant colocalize with MRC-1. Hence, our data suggests that PLY binding to MRC-1 occurs independently of binding to other ligands such as capsular polysaccharides.

Figure 1 adapted from (Subramanian *et al.*, 2019). Plate-bound recombinant MRC-1 (5µg/ml) was incubated with full-length pneumolysin (PLY), purified type 2 and type 4 capsules or PLY+Capsule for 1 hr at 37°C. BSA was used as a negative control for the binding assay. Bound PLY and capsule was detected using antibodies

5) Fig 3E shows a very low-resolution study of the ability of the toxin to be neutralized by the peptide. A higher resolution study focusing on the dose response to inhibit hemolysis by pure toxin vs peptide would be important.

Author Response: Thank you for the comment. We have in Revised Figs. 3B and EV3C and D, performed a dose-response curve showing inhibition of the purified toxins with the peptides, P2 and P3, and control peptides.

6) Specifically to show potency which is measured by the dose where the hemolysis begins to be affected, the slope of the inhibition to determine the MD50 and the plateau at full effect to show neutralization can be 100%.

Author Response: According to the suggestion by the reviewer, we have now determined the potency of the peptides against the purified toxins by calculating the median effective dose, ED50 using non-linear regression analysis (curve fitting). The ED50 values of the peptides against the purified toxins are shown in the table below. This table is now included as Appendix Table S3 in the revised manuscript and described in the text (Page 7, lines 19-23).

Appendix Table S3. ED50 values of peptides P2 and P3 vs the purified CDC toxins.

Peptide ED50 (μ M)	PLY	LLO	SLO
P2	6.6	22.3	9.8
P3	21.9	88.5	17.7

Referee #2 (Remarks for Author):

This interesting study by Subramanian et al. builds on the previous identification of MRC-1 as an immune cell receptor for pneumolysin (Ply). The authors identify peptides derived from MRC-1 that bind to Ply, as well as to other related toxins such as LLO and SLO. These peptides diminish the activities of Ply, such as cytolysis, proinflammatory cytokine production, epithelial integrity and MRC-1-mediated internalization. These peptides, like genetic ablation of ply, resulted in targeting of bacteria to autophagosomes and enhanced bacterial killing. The linkage of a peptide to nanoparticles resulted in a reagent that increased bacterial clearance and host survival in zebrafish and mouse models of infection, albeit with relatively modest changes in disease outcome.

Author Response: We thank the reviewer for the positive comment!

Major comments:

1. Specificity of inhibitory peptides. (a) Fig. S2C shows that all MRC-1 peptides appear to have some inhibitory effect on hemolysis. How do the authors explain the inhibitory effect on MRC-1 peptides that do not interact with the loop?

Author Response: Thank you for this question. We would like to clarify that all the 6 peptides were derived from the CTLD4 domain of MRC-1 that interacts with the cholesterol binding loop of the CDCs and possessed at least one residue forming hydrogen bonding interaction. Hence, there is a general effect with peptides P1-6, but not with the control peptides, which had no activity. However, specifically peptides P2 and P3 had 5 and 3 amino

acids respectively in a continuous stretch that formed hydrogen-bonding interactions with the loop. In comparison, peptides P1 had 2 interspersed residues, while peptides P4, 5 and 6 had only one residue that formed H-bonding with the CDC loop (Appendix Table S1). The residues involved in hydrogen bonding are listed in Appendix Table S2. This may explain the higher activity of P2 and P3 compared to the other peptides. We have now performed an ANOVA test to compare P2 and P3 with the other peptides and found that the difference is significant (Fig. EV2C in the revised manuscript). We have now added text in the revised manuscript to clarify this (Page 6, line 25 and page 7 lines 1-2).

(b) Note that the peptides each have more acidic than basic residues; one of the two control peptides shares this property. A superior control peptide would be a 'scrambled' version of P2 or P3, depending on which peptide is being tested. Such a peptide need not be tested in all of the assays described, but a few key assays that demonstrated that the peptide sequence rather than just its amino acid composition.

Author Response: Thank you for the excellent suggestion. Accordingly, we have now tested a scrambled version of peptide P2 (PDSTFWNGESVYS) in inhibiting toxin-induced hemolysis (Figure 2, below) and LDH-release assay in THP-1 macrophages (Figure 3, below). We found that while P2 and P3 showed dose-dependent reduction in hemolytic activity of toxins, scrambled P2 did not have any significant effect on hemolysis induced by the toxins, PLY, LLO or SLO (Figure 2, below). Figure 2A, 2B and 2C below are incorporated as Fig. 3B, Fig. EV3C, EV3D respectively in the revised manuscript and described in the text (Page 7, lines 16-21). Further, we also performed LDH-cytotoxicity assays in THP-1 macrophages stimulated with purified toxins with or without peptides. In agreement with the data from the hemolysis assay, P2 and P3 significantly reduced cell death of macrophages induced by PLY, LLO or SLO (Figure 3, below), but no significant effect was observed with the controls, scrambled P2 or CP2. Figure 3 below is incorporated as Fig. 3C in the revised manuscript and described in the text (Page 8, lines 17-18). The above results confirm that the peptide sequence rather than its amino acid composition is crucial for inhibiting toxin activity.

Figure 2. Hemolysis assay using purified (A) PLY, (B) LLO and (C) SLO in the presence of increasing concentrations of MRC-1 peptides, P2, P3 and control peptides, scrambled P2 and CP2(1-1000 μ M). * denotes $P < 0.05$ by one-way ANOVA with Dunnett's post-test. n.s. denotes not significant.

Figure 3. LDH cytotoxicity assay in human THP-1 macrophages stimulated with purified PLY, LLO or SLO (0.5µg/ml) in the presence or absence of 100 µM peptides P2, P3 or control peptides, scrambled P2 and CP2 for 18h. Cholesterol (100 µM) was used as positive control to inhibit hemolysis. Data are mean ± s.e.m from 4 independent experiments. **** denotes $P < 0.0001$ by two-way ANOVA with Bonferroni post-test. n.s. denotes not significant.

(c) The authors choose to focus on two, i.e. P2 and P3. A statistical analysis of P2 and P3 vs. other peptides should be performed.

Author Response: According to the suggestion by the referee, we have now performed statistical analysis of P2 and P3 vs other peptides using one-way ANOVA with Tukey's post test. The modified figure with statistics is included in the revised manuscript as Fig. EV2C.

(d) Fig. 3D. What explains the surprising punctate staining pattern of Sp by P2? Is it dependent on capsule, i.e. does the acapsular mutant give the same pattern?

Author Response: Following the suggestion, we have now tested binding of FITC-labelled P2 (green) to the capsular mutant type 4 strain, T4R (Red), by immunofluorescence microscopy (Figure 4, below). We found that P2 also binds to T4R predominantly in a punctate staining pattern. The yellow indicates the regions of colocalization with T4R. No binding was observed with the control peptide, CP2, confirming the binding specificity (Figure 4B). In agreement with our findings, a study by Shak et al. also found that PLY is localized to the pneumococcal cell wall in a punctate pattern using electron microscopy (Shak *et al*, 2013).

Figure 4. Binding of FITC-labelled peptide P2 to the capsular mutant strain T4R. Binding of (a) FITC labelled P2 and (b) control peptide CP2, to Nile-red stained T4R was visualized using fluorescence microscopy. Yellow indicates colocalization of peptide with Nile-red. Scale bars, 5 µm.

Is the same staining pattern also revealed using anti-Ply antibody?

Author Response: Following the reviewer's suggestion, we have now stained the encapsulated T4 strain using anti-Ply antibody (Figure 5, below). In agreement with staining of T4 by P2, anti-Ply staining (green) also revealed a predominantly punctate staining pattern. The PLY mutant strain, T4 Δ ply, was used as a negative control (Figure. 5B below).

Figure 5. Anti-Ply staining of wild-type T4 pneumococci. Immunofluorescence microscopy showing the staining pattern of (a) wild-type T4 pneumococci using anti-Ply and Alexa 488 conjugated secondary antibody. (b) The PLY deficient strain, T4 Δ ply, was used as a negative control. Scale bars, 10 μ m.

2. P. 4, line 7. Does co-localization of LLO and SLO with EEA-1 require MRC-1, as they have shown previously for PLY utilizing knock-down of MRC-1?

Author Response: In line with the reviewer's suggestions, we have now blocked MRC-1 in DCs using antibodies and tested the binding and colocalization of LLO and SLO with EEA-1 (Figure 6, below). We found upon MRC-1 blockade, that the binding of LLO and SLO to DCs was much reduced (Figure 6B, D below) when compared to no blockade (Figure 6A, C below). The basal binding could be due to interactions with membrane cholesterol. Further, colocalization of LLO and SLO with the endosomal marker EEA-1 was significantly reduced upon MRC-1 blockade. Figure 6B, and D below have been included in the revised manuscript as Fig EV1D and E and described in the text (page 5, lines 16-18). Further, we have also quantified the extent of colocalization using the Pearson's correlation coefficient in the revised manuscript (Fig. EV1F).

Figure 6. Effect of MRC-1 antibody blockade on binding and colocalization of LLO and SLO with EEA-1 in DCs. Primary human DCs were incubated with 200 ng/ml of purified (A, B) LLO and (C, D) SLO for 60 min with or without pretreatment with 1 μ g/ml of anti-MRC-1. Immunofluorescence microscopy shows that upon MRC-1 blockade the binding of LLO and SLO (green) to DCs is diminished and they do not colocalize with the early endosomal antigen, EEA-1 (pink). Scale bars, 10 μ m.

3. Explanation of Ply concentrations utilized, a particularly important issue that this group addressed in a previous publication. (a) P. 7, 5th line from bottom. Concentration of Ply utilized was 500 µg/ml, or 2.5 x the concentration typically utilized in previous publication. Can the authors provide an explanation of how concentration was chosen, given the concentration-dependent response to CDC?

Author Response: In the previous study, the aim was to investigate the immunomodulatory effect of PLY at sublytic doses (≤ 0.2 µg/ml) on immune cells. In the current study, we wanted to test the efficacy of the peptides to inhibit cytolysis induced by PLY and related CDCs. In the literature, Ply concentrations ranging from 0.5-1 µg/ml have been shown to activate inflammasome and induce acute lung injury in acute pneumonia model (Shoma *et al*, 2008) (Witzenrath *et al.*, 2006) (McNeela *et al*, 2010). Besides, in rabbit models of pneumococcal meningitis, concentrations of PLY up to 4.34 µg/ml has been measured in the cerebrospinal fluid (Stringaris *et al*, 2002). Hence, in this study, we used higher doses of 0.5-1 µg/ml in order to mimic pneumolysin induced cytolysis and inflammation. In the revised manuscript, we have now described the rationale for choosing the concentrations (Page 8, lines 4-7).

(b) P. 9. middle of page. What is final concentration when 50 µg are added to tissue model? The authors should describe rationale for choosing this concentration.

Author Response: Thank you for pointing this out. The dosage used was 50 ng in 50 µl at a concentration of 1 µg/ml. To be consistent, we have now mentioned the concentration in the revised manuscript (page 9, line 22). This was optimized based on previous experience with the sensitivity of the lung epithelial tissue model (Mairpady Shambat *et al*, 2015) to bacterial toxins where 0.45-0.9 µg/ml of toxin was used. Two doses of Ply (0.5 and 1 µg/ml) were tested preliminarily and 1 µg/ml Ply induced significant epithelial damage in the models. Hence, 1 µg/ml was chosen for experiments with the lung epithelial tissue model system.

(c) Fig. 3C and 4A. The concentration of Ply used when measuring cytokine production causes 90% cytotoxicity. How might this alter the production of cytokines?

Author Response: The objective of these experiments was to study the potential inhibitory effect of peptides on PLY-induced lytic cell death of macrophages (Fig 3C) and ensuing cytokine release (Fig. 4A). At concentrations of 0.5 µg/ml and above, Ply has been shown to induce activation of NLRP3 inflammasome and pro-inflammatory cytokine release by immune cells (McNeela *et al.*, 2010). This is dependent on the cytolytic activity of PLY. Hence, we used this concentration (0.5 µg/ml) to mimic Ply-induced cytolysis and cell death-associated cytokine release.

4. Fig. 4D and E. (a) Can the authors rule out the possibility that it is bacterial intracellular survival that is diminished by P2 or by the Ply-deficient strain? (b) It is not clear how efficient bacterial invasion is. The authors should convert CFU/ml into % entry to convey this information.

Author Response: Thank you for this suggestion. To clarify, both bacterial invasion (Fig. 4D) as well as intracellular survival (Fig. 4E) are diminished by P2. As suggested, we have now converted CFU/ml into % entry in the revised manuscript for clarification (Figure 7, below). The % bacterial entry was calculated using the equation below
%bacterial entry= (Bacteria uptaken at 2h/Input)x100.

Figure 7 below is included as Fig. 4D in the revised manuscript.

Figure 7. % Bacterial entry of wild-type pneumococci T4 (TIGR4) or its isogenic PLY mutant T4 Δ ply into the lung epithelial models (n=3/condition) in the presence or absence of peptide P2 or CP2. % entry was calculated using the formula (Bacterial uptaken at 2h/Input) \times 100. Anti-PLY was used as control to test the effect of blocking PLY. Data in d and e are mean \pm S.E.M. of n=3 models/condition from two independent experiments. ** denotes P < 0.005 by one-way ANOVA with Dunnet's post-test. n.s. denotes not significant.

5. P. 11, bottom and Fig. 4F and G. How do the authors reconcile their findings with those of Omnishi and coworkers (Cell Reports '20 and Autophagy '20) and others who report that Ply triggers autophagy in a variety of cell types?

Author Response: We would like to clarify that in the studies by Omnishi et al. (Cell Reports '20 and Autophagy '20), they have used human embryonic kidney cells and mouse embryonic fibroblasts which do not express the human mannose receptor C type lectin receptor (MRC-1) and are fundamentally different cell types compared to the human dendritic cells used in Fig. 4F and G. MRC-1 is specifically expressed on dendritic cells and tissue macrophages such as alveolar macrophages in the lungs. In our earlier study, we have shown that pneumolysin elicits response in a cell-type specific manner (Subramanian *et al.*, 2019), and hence this could explain the difference in the results of the studies.

6. The panels comparing T4R vs. T4R delta ply are difficult to compare because color of LC3B is different in the two panels.

Author Response: Thank you for the suggestion. Accordingly, we have now made the color of LC3B uniform in both the figures. For more clarity, we have also added the quantification data showing the extent of colocalization in the revised manuscript (Fig. 4G). The panel showing the negative control of T4R Δ ply infected DCs is included as Fig. EV5A-C in the revised manuscript.

7. Fig. 5. Can the authors rule out that the greater efficacy of P2-NPs compared to NP's or P2 alone is due to the agglutination of bacteria by the beads rather than specific inhibition of PLY activity?

Author Response: We would like to clarify that both the P2-NPs as well as the NPs alone control had the nanoparticles. Hence, this would rule out any potential side-effects of the NPs alone on the bacteria. Further, as another control, we tested NPs loaded with the control peptide (CP2-NPs) in the bacterial hemolysis assay (Appendix S2D). We found that in

contrast to P2-NPs, CP2-NPs did not inhibit hemolysis of wild-type T4 strain expressing PLY, confirming the specificity of P2-NPs and ruling out any effects of agglutination by the NPs.

Minor points:

8. P. 6, last line. the authors should describe how they determined which residues interact with the loop 4. (It is clear that they used modeling, but what were the criteria for assigning interacting residues?)

Author Response: The interacting residues indicated were the amino acids that formed hydrogen bonding interactions with the cholesterol binding loop residues. The amino acids involved in hydrogen bonding are listed in Appendix Table S2. We have now also described this in the revised manuscript text (Page 7, lines 1-2).

9. P. 6. line 7. The "cholesterol binding loop" alters the avidity of cholesterol binding by altering oligomerization, rather than directly binding cholesterol (see Dowd and Tweten, PLoS Pathogen, 2012); line 12 and elsewhere should also be edited accordingly.

Author Response: As suggested, we have now edited the statement in Page 6, lines 5-6 and cited the article by Dowd and Tweten, PLoS Pathogen, 2012.

10. P. 6, line 4. "show" should read "suggest"

Author Response: We have now modified the sentence accordingly in the revised manuscript text (Page 6 lines 6-7).

11. P. 6, line 6. "we found that tryptophan" should read "modeling predicted that tryptophan"

Author Response: We have now modified the sentence accordingly (Page 6, line 8) in the revised manuscript.

12. Fig. 5B and C are difficult to discern the different groups.

Author Response: Thank you for pointing this out. For clarity, we have now labelled the groups adjacent to the curves in Figs. 5B and C.

Referee #3 (Remarks for Author):

Very elegant work showing that peptide binding/masking of the pneumococcal toxin pneumolysin influences the intracellular fate of the bacteria. Blocking of the ply dependent inhibition of phagosome maturation led to successful autophagy of pneumococci, which in the presence of active ply could survive in the cells.

Author Response: We thank the reviewer for the positive comment!

Main comments:

Figure 1 shows very convincingly colocalization of PLY and LLO with MRC-1 and EEA-1, but SLO appears to colocalise with MRC-1, but not EEA-1. Showing a bar chart showing the extent of colocalization would solve the issue and allow to clearly report the results in the text.

Author Response: Thanks for acknowledging that the colocalization data is convincing, and also for the excellent suggestion. Following the suggestion, we have now included a graph quantifying the extent of colocalization of MRC-1 with PLY, LLO and SLO by calculating the Pearson's correlation coefficient (Figure 8 below). The data below have been included in the revised manuscript as Figs. EV1 A and F respectively and mentioned in the revised text (Page 5, lines 9-10 and 17-18).

Figure 8. Extent of colocalization of PLY, LLO and SLO with MRC-1 and EEA-1 in DCs. Graphs showing the Pearson's correlation coefficient to measure colocalization of PLY, LLO, SLO with (a) MRC-1 and (b) EEA-1 in primary human DCs. Upon antibody blockade of MRC-1, the extent of colocalization with EEA-1 was much reduced. * $P < 0.05$; ** $P < 0.01$ by two-way ANOVA with Bonferroni post-test.

When comparing panels in Figure 4F to those of 4G, it would appear that there is much more red stain (mannose receptor) in the left panels and much more pink stain in the right panels, as if the interaction with ply could somehow have upregulated mannose receptor and/or downregulated LC3B. As such an effect could be caused by indirect effects or cause downstream indirect effects, it would be important if authors could quantify the reactivity in the images/samples and confirm that the markers are equally expressed or distributed.

Author Response: Thank you for this comment. We would like to clarify that MRC-1 expression is indeed upregulated in cells infected by T4R as compared to T4RΔply. We have already reported this in our previous study (Subramanian *et al.*, 2019).

To measure the activation of the autophagy marker, LC3B, we have now performed western blotting to quantify LC3B expression in DCs infected with T4R or the PLY mutant, T4RΔply. Upon induction of autophagy, pro-LC3 is proteolytically cleaved by ATG4 into LC3-I and subsequently conjugated to phosphoethanolamine to generate the LC3-II version having higher mobility on SDS-PAGE (Runwal *et al.*, 2019). We found that expression of both LC3B-I and II is higher in DCs infected with T4RΔply when compared to T4R infected cells (Figure 9 below). Also, blockade of PLY-MRC1 interaction by P2 induced higher expression of LC3B-I and LC3B-II. This data agrees with our microscopy images shown in Fig. 4F, G (Figs. 4F and EV5 respectively in the revised manuscript) and explains why the signal for LC3B is higher in DCs infected with T4RΔply. Hence, the data suggests that pneumococci may inhibit maturation of autophagosome in DCs in a PLY-dependent manner to promote

intracellular survival. In support of this, the intracellular pathogen, mycobacterium, has been shown to inhibit autophagy to survive within macrophages (Chandra *et al*, 2015).

For additional clarity, we have included quantification data showing the extent of colocalization of bacteria with both MRC-1 and LC3B in cells infected with T4R and T4R Δ ply (Fig. 4G in the revised manuscript).

Figure 9. LC3B activation in infected DCs upon treatment with MRC-1 peptides.

Western blotting showing the levels of LC3B-I and the activated form, LC3B-II in DCs treated with 100 μ M of peptide P2 or control peptide CP2 and infected with T4R (MOI of 10) for 2h. The PLY mutant, T4R Δ ply, was used as a control. Cells treated with 50 μ M of the autophagy inducer chloroquine was used as the positive control.

Minor comments:

The scale ticks and labelling of the X-axis of Fig 3E appear to be wrong ones?

Author Response: Thank you for noticing this. We have now corrected the X- axis labelling in Fig. 3E.

References cited in this response letter:

Chandra P, Ghanwat S, Matta SK, Yadav SS, Mehta M, Siddiqui Z, Singh A, Kumar D (2015) Mycobacterium tuberculosis Inhibits RAB7 Recruitment to Selectively Modulate Autophagy Flux in Macrophages. *Scientific reports* 5: 16320

Garcia-Suarez Mdel M, Cima-Cabal MD, Florez N, Garcia P, Cernuda-Cernuda R, Astudillo A, Vazquez F, De los Toyos JR, Mendez FJ (2004) Protection against pneumococcal pneumonia in mice by monoclonal antibodies to pneumolysin. *Infection and immunity* 72: 4534-4540

Garcia-Suarez Mdel M, Florez N, Astudillo A, Vazquez F, Villaverde R, Fabrizio K, Pirofski LA, Mendez FJ (2007) The role of pneumolysin in mediating lung damage in a lethal pneumococcal pneumonia murine model. *Respir Res* 8: 3

Humar D, Datta V, Bast DJ, Beall B, De Azavedo JC, Nizet V (2002) Streptolysin S and necrotising infections produced by group G streptococcus. *Lancet* 359: 124-129

Mairpady Shambat S, Chen P, Nguyen Hoang AT, Bergsten H, Vandenesch F, Siemens N, Lina G, Monk IR, Foster TJ, Arakere G *et al* (2015) Modelling staphylococcal pneumonia in a human 3D lung tissue model system delineates toxin-mediated pathology. *Disease models & mechanisms* 8: 1413-1425

McNeela EA, Burke A, Neill DR, Baxter C, Fernandes VE, Ferreira D, Smeaton S, El-Rachkidy R, McLoughlin RM, Mori A *et al* (2010) Pneumolysin activates the NLRP3 inflammasome and promotes proinflammatory cytokines independently of TLR4. *PLoS pathogens* 6: e1001191

Runwal G, Stamatakou E, Siddiqi FH, Puri C, Zhu Y, Rubinsztein DC (2019) LC3-positive structures are prominent in autophagy-deficient cells. *Scientific reports* 9: 10147

Shak JR, Ludewick HP, Howery KE, Sakai F, Yi H, Harvey RM, Paton JC, Klugman KP, Vidal JE (2013) Novel role for the *Streptococcus pneumoniae* toxin pneumolysin in the assembly of biofilms. *mBio* 4: e00655-00613

Shoma S, Tsuchiya K, Kawamura I, Nomura T, Hara H, Uchiyama R, Daim S, Mitsuyama M (2008) Critical involvement of pneumolysin in production of interleukin-1 α and caspase-1-dependent cytokines in infection with *Streptococcus pneumoniae* in vitro: a novel function of pneumolysin in caspase-1 activation. *Infection and immunity* 76: 1547-1557

Stringaris AK, Geisenhainer J, Bergmann F, Balshusemann C, Lee U, Zysk G, Mitchell TJ, Keller BU, Kuhnt U, Gerber J *et al* (2002) Neurotoxicity of pneumolysin, a major pneumococcal virulence factor, involves calcium influx and depends on activation of p38 mitogen-activated protein kinase. *Neurobiol Dis* 11: 355-368

Subramanian K, Neill DR, Malak HA, Spelmink L, Khandaker S, Dalla Libera Marchiori G, Dearing E, Kirby A, Yang M, Achour A *et al* (2019) Pneumolysin binds to the mannose receptor C type 1 (MRC-1) leading to anti-inflammatory responses and enhanced pneumococcal survival. *Nature microbiology* 4: 62-70

Witzenrath M, Gutbier B, Hocke AC, Schmeck B, Hippenstiel S, Berger K, Mitchell TJ, de los Toyos JR, Rosseau S, Suttorp N *et al* (2006) Role of pneumolysin for the development of acute lung injury in pneumococcal pneumonia. *Critical care medicine* 34: 1947-1954

26th Aug 2020

Dear Prof. Henriques-Normark,

Thank you for the submission of your revised manuscript to EMBO Molecular Medicine. We have now received the enclosed reports from the referees that were asked to re-assess it. As you will see the reviewers are now globally supportive and I am pleased to inform you that we will be able to accept your manuscript pending the following final amendments:

***** Reviewer's comments *****

Referee #1 (Comments on Novelty/Model System for Author):

The manuscript has been extensively revised and the authors have strongly addressed the reviewers comments.

Referee #1 (Remarks for Author):

extensive revisions address reviewers comments

Referee #2 (Comments on Novelty/Model System for Author):

Very much improved manuscript.

Referee #2 (Remarks for Author):

This interesting study by Subramanian et al. has been significantly strengthened by the authors' systematic additions of new experiments and clarified text. The format of the point-by-point response was very helpful. The authors should address the following points:

1. Fig. 4D and E. The authors distinguish entry and survival by performing gentamicin protection experiments in their 3D lung epithelial model or in human DC's. Given that the same assay is used in both experiments, it is difficult to distinguish the two processes by these experiments alone, i.e. in both assays, differences in entry or survival (or both) could account for their findings in both panels. The authors could simply state the caveat that these assays don't distinguish the processes (which I don't think is a major point of their study). Alternatively, they could perform a time course of survival that would determine if survival in human DCs is indeed compromised by P2. (See Novakowski et al., Chapt 20 in Roberto Botelho (ed.), Phagocytosis and Phagosomes: Methods and Protocols, Methods in Molecular Biology, vol. 1519, DOI 10.1007/978-1-4939-6581-6_20, Springer

Science/Business Media New York 2017).

2. The previous question as to whether the authors can rule out that the greater efficacy of P2-NPs compared to NP's or P2 alone is due to the agglutination of bacteria by the beads rather than specific inhibition of PLY activity is not really addressed by the experiments cited in the response, in that my original question is whether nanoparticles coated with P2 might have a biological effect simply by attaching to bacteria as opposed to attaching to and neutralizing an activity of PLY. This would require comparison to particles coated with anti-capsule or some other agent that binds the bacterial surface. I think it sufficient to instead simply state the caveat that the experiments cannot rule out this possibility.

Reviewer comments**Reviewer #1**

(Comments on Novelty/Model System for Author):

The manuscript has been extensively revised and the authors have strongly addressed the reviewers comments.

(Remarks for Author):

extensive revisions address reviewers comments

Response: We thank the reviewer for the positive comments.

Reviewer #2

(Comments on Novelty/Model System for Author):

Very much improved manuscript.

Response: We thank the reviewer for the positive comment.

(RemarksforAuthor):

This interesting study by Subramanian et al. has been significantly strengthened by the authors' systematic additions of new experiments and clarified text. The format of the point-by-point response was very helpful.

Response: We thank the reviewer for the positive comments

The authors should address the following points:

1. Fig. 4D and E. The authors distinguish entry and survival by performing gentamicin protection experiments in their 3D lung epithelial model or in human DC's. Given that the same assay is used in both experiments, it is difficult to distinguish the two processes by these experiments alone, i.e. in both assays, differences in entry or survival (or both) could account for their findings in both panels. The authors could simply state the caveat that these assays don't distinguish the processes (which I don't think is a major point of their study). Alternatively, they could perform a time course of survival that would determine if survival in human DCs is indeed compromised by P2. (See Novakowski et al., Chapt 20 in Roberto Botelho (ed.), Phagocytosis and Phagosomes: Methods and Protocols, Methods in Molecular Biology, vol. 1519, DOI 10.1007/978-1-4939-6581-6_20, Springer Science/Business Media New York 2017).

Response: According to the reviewer's suggestion, we have now included the statement below in the revised manuscript to indicate that the assays don't necessarily distinguish entry and survival (Page 12, lines 13-14).

“The inhibition of bacterial invasion by the peptides into the lung epithelium and DCs (Fig. 4D and 4E) could be due to reduced bacterial entry as well as intracellular survival.”

2. The previous question as to whether the authors can rule out that the greater efficacy of P2-NPs compared to NP's or P2 alone is due to the agglutination of bacteria by the beads rather than specific inhibition of PLY activity is not really addressed by the experiments cited in the response, in that my original question is whether nanoparticles coated with P2 might have a biological effect simply by attaching to bacteria as opposed to attaching to and neutralizing an activity of PLY. This would require comparison to particles coated with anti-capsule or some other agent that binds the bacterial surface. I think it sufficient to instead simply state the caveat that the experiments cannot rule out this possibility.

Response: According to the reviewer's suggestion, we have now included the below statement in the revised manuscript (page 17, lines 18-19) to indicate the possibility of effects due to binding of P2-coated NPs to the bacterial surface.

“Further, any possible effects due to interaction of NPs with bacteria are not completely ruled out and needs further studies”.

The authors performed the requested changes.

Corresponding Author Name: Birgitta Henriques-Normark

Manuscript Number: EMM-2020-12695